# POT1a deficiency in mesenchymal niches perturbs B-lymphopoiesis

Kentaro Nakashima[1,2], Yuya Kunisaki [1,3,4✉], Kentaro Hosokawa[1], Kazuhito Gotoh [5], Hisayuki Yao[1], Ryosuke Yuta[1], Yuichiro Semba[3,4], Jumpei Nogami[3,4], Yoshikane Kikushige[3,4], Patrick S. Stumpf [6], Ben D. MacArthur[7,8,9], Dongchon Kang[5], Koichi Akashi[4], Shouichi Ohga[2] & Fumio Arai [1✉]

Protection of telomeres 1a (POT1a) is a telomere binding protein. A decrease of POT1a is related to myeloid-skewed haematopoiesis with ageing, suggesting that protection of telomeres is essential to sustain multi-potency. Since mesenchymal stem cells (MSCs) are a constituent of the hematopoietic niche in bone marrow, their dysfunction is associated with haematopoietic failure. However, the importance of telomere protection in MSCs has yet to be elucidated. Here, we show that genetic deletion of POT1a in MSCs leads to intracellular accumulation of fatty acids and excessive ROS and DNA damage, resulting in impaired osteogenic-differentiation. Furthermore, MSC-specific POT1a deficient mice exhibited skeletal retardation due to reduction of IL-7 producing bone lining osteoblasts. Single-cell gene expression profiling of bone marrow from POT1a deficient mice revealed that B-lymphopoiesis was selectively impaired. These results demonstrate that bone marrow microenvironments composed of POT1a deficient MSCs fail to support B-lymphopoiesis, which may underpin age-related myeloid-bias in haematopoiesis.

[1] Department of Stem Cell Biology and Medicine, Graduate School of Medical Sciences, Kyushu University, Fukuoka, Japan. [2] Department of Pediatrics, Graduate School of Medical Sciences, Kyushu University, Fukuoka, Japan. [3] Center for Cellular and Molecular Medicine, Kyushu University Hospital, Fukuoka, Japan. [4] Department of Medicine and Biosystemic Science, Kyushu University Graduate School of Medical Sciences, Fukuoka, Japan. [5] Department of Clinical Chemistry and Laboratory Medicine, Graduate School of Medical Sciences, Kyushu University, Fukuoka, Japan. [6] Joint Research Center for Computational Biomedicine, RWTH Aachen University, Aachen, Germany. [7] Centre for Human Development, Stem Cells and Regeneration, Faculty of Medicine, University of Southampton, Southampton, UK. [8] Mathematical Sciences, University of Southampton, Southampton, UK. [9] The Alan Turing Institute, London, UK. ✉email: kunisaki.yuya.519@m.kyushu-u.ac.jp; arai.fumio.603@m.kyushu-u.ac.jp

Telomeres are chromosomal 3′-terminal structures consisting of specific tandem repeats, 5′-TTAGGG −3′ that protect chromosomes from DNA degradation. Shelterin is a protein complex composed of six subunit proteins, TRF1, TRF2, TIN2, TPP1, RAP1 and Protection of telomere 1 (POT1), which maintains telomere length and genomic integrity[1,2]. The shelterin complex is also reported to exhibit a DNA remodelling activity by recruiting DNA-repair factors to change the structure of telomeric DNA, thereby protecting chromosome ends[3]. POT1 is a protein that binds to a single-stranded TTAGGG repeat through its direct interaction with TPP1. There are two POT1 orthologs, POT1a and POT1b in the mouse genome, which have different functions in telomeres: POT1a represses DNA damage signals, and POT1b regulates the amount of single-stranded DNA at the telomere terminus, respectively[4,5]. Previous studies with genetic mouse models report that POT1a deficient mice exhibit early embryonic lethality, whereas POT1b deficiency exhibits a dyskeratosis congenita-like phenotype only in a telomerase-haploinsufficient background, suggesting that POT1a is functionally dominant in development[5,6].

We previously reported that POT1a prevents hematopoietic stem cells (HSCs) from being skewed to myeloid commitment, which occurs during ageing, by protecting HSCs from DNA damage responses (DDRs) and reactive oxygen species (ROS) at telomeres[7], suggesting that protection of telomeres is important for stem cells to sustain their self-renewal and multi-lineage differentiation potential for life.

Mesenchymal stem cells and multipotent stromal cells (MSCs) are characterised by their ability to self-renew and differentiate into osteogenic-, chondrogenic- and adipogenic- lineages. Cells exhibiting an MSC activity reside in bone marrow (BM) in human and mice. Upon injury, bone marrow MSCs contribute to the regeneration of bone, cartilage and fat tissue through proliferation and differentiation and exhibit anti-inflammatory effects by secreting cytokines[8]. Although human MSCs share the characteristics with murine ones, they are isolated with some unique surface markers other than the orthologs. In addition, human MSCs have been elucidated to play roles in various pathologies due to their predisposition to engraft in xenograft mouse models[9–11]. MSCs are a heterogeneous subset of stromal stem cells that can be isolated from many tissues. Some of the subpopulations of MSCs can be isolated from BM using a panel of surface markers. In BM, MSCs have been shown to line the vascular endothelium and form a local microenvironment referred to as the *niche*, which supports haematopoiesis including the maintenance of HSCs that can produce all mature blood cell lineages[12]. In the mouse BM of Nestin-GFP transgenic mice, where GFP is expressed under the control of the endogenous Nestin promoter, all of the MSCs' differentiation capacity is found in Nestin-GFP+ cells, ~98% of which are marked by NG2-cre[13,14].

BM microenvironments are remodelled with ageing, which can be triggered by the loss of β3 adrenergic receptor signalling caused by a reduced number of adrenergic nerves associated arterioles in BM[15,16]. In the age-remodelled BM, MSCs show alteration that reduced production of niche factors that support HSC maintenance, increased adipogenesis and reduced osteogenesis which contribute to a myeloid skewed phenotype of HSCs in aged marrow[17]. Several studies show that microenvironments composed of aged MSCs associate with leukemia development[18]. However, whether cell intrinsic ageing of MSCs affects their function to support normal and pathogenic haematopoiesis, or as a source of tissue regeneration, remains unknown.

In this study, we investigate the influence of telomere dysfunction in BM MSCs in vitro and in vivo via genetic deletion of POT1a. We show that POT1a is crucial for osteogenic-differentiation from MSCs, and we find that the ability to support B-lymphopoiesis is diminished in BM microenvironments composed of MSCs with POT1a deletion.

## Results

**POT1a deletion in MSCs leads to impaired osteogenic-lineage differentiation**. BM microenvironments are remodelled with ageing, which is closely related to a decline of haematopoiesis and development of blood and immune disorders. The function of MSCs in HSC niches is reported to decline with ageing. First, to evaluate age-related changes to the differentiation potential of BM MSCs, we analysed the expression of genes associated with osteogenic-, adipogenic- and chondrogenic- differentiation in the BM PDGFRα+ CD51+ MSCs, which mark human Nestin+ sphere-forming MSCs capable of hematopoietic progenitor cell expansion, sorted from 8-10 weeks old adult and ~120 weeks old aged mice[19]. The aged mice exhibited significantly lower expression of genes associated with osteogenic differentiation compared to adult mice (Fig. 1a). As alteration of the length and function of telomeres are associated with a decline of cellular and metabolic homeostasis in the processes of ageing, we compared the length of telomeres and the expression of molecules essential for protection of telomeres in the MSCs from adult and aged mice. Although the length of telomeres was comparable, POT1a was significantly decreased in the aged MSCs (Fig. 1b, c). Moreover, we have performed experiments, in which MSCs were transfected with a retroviral vector to overexpress POT1a in culture. The rescue with POT1a in MSCs isolated from the old mice has restored the expression of genes related to osteogenesis, such as *Alpl*, *Col1a1*, and *Opn*, suggesting the importance of POT1a for MSCs to sustain multi-potency (Fig. 1a).

These data prompted us to investigate roles of POT1a in BM MSCs. To this end, we performed assays with BM PDGFRα+ CD51+ MSCs isolated from Pot1a^flox/flox^/flox-stop-flox tdTomato mice, in which deletion of POT1a can be induced by administration of Cre recombinant vesicles in vitro, denoted Pot1a^Δ/Δ^ MSCs (Fig. 1d and Supplementary Fig. 1a). Genetic deletion of POT1a was confirmed to give a significant reduction (~90%) of mRNA, yet no significant effect on the telomere length or proliferation of the MSCs was observed (Fig. 1d–g and Supplementary Fig. 1b). To examine comprehensively the effects of POT1a deficiency on MSC functions, we performed RNA-sequencing analyses of the Pot1a^Δ/Δ^ MSCs in vitro. MSCs with POT1a deletion exhibited distinct gene expression profiles compared to POT1a wildtype, 4 weeks after Cre administration (Fig. 1h, i). Gene set enrichment analysis (GSEA) of the differentially expressed genes revealed that several pathways related to bone development were significantly affected (Fig. 1j and Supplementary Fig. 1c). Quantitative PCR (qPCR) analyses of the MSCs after POT1a deletion showed significant decreases in expression of genes associated with osteogenic- (*Alpl*, and *Opn*) differentiation while expression of genes associated with chondro- (*Sox9*) and adipo- (*Pparγ*) differentiation were not significantly affected. Expression of most of the genes encoding factors to support HSC activity (HSC niche-related) (*CXCL12, KitL, Vcam1* and *Angpt1*) and multipotency (*Nanog* and *Sox2*) were not significantly reduced. These results suggest that reduced multi-potency may be the primary effect of POT1a deletion (Fig. 1k and Supplementary Fig. 1d–f). To confirm these results, we performed in vitro differentiation assays, in which MSCs with and without POT1a deletion were cultured in osteogenic, adipogenic and chondrogenic differentiation media, respectively. POT1a-deficient MSCs failed to differentiate into osteocytes and chondrocytes efficiently, while adipocyte differentiation was comparable to POT1a wildtype (Fig. 1l and Supplementary Fig. 1g). These results

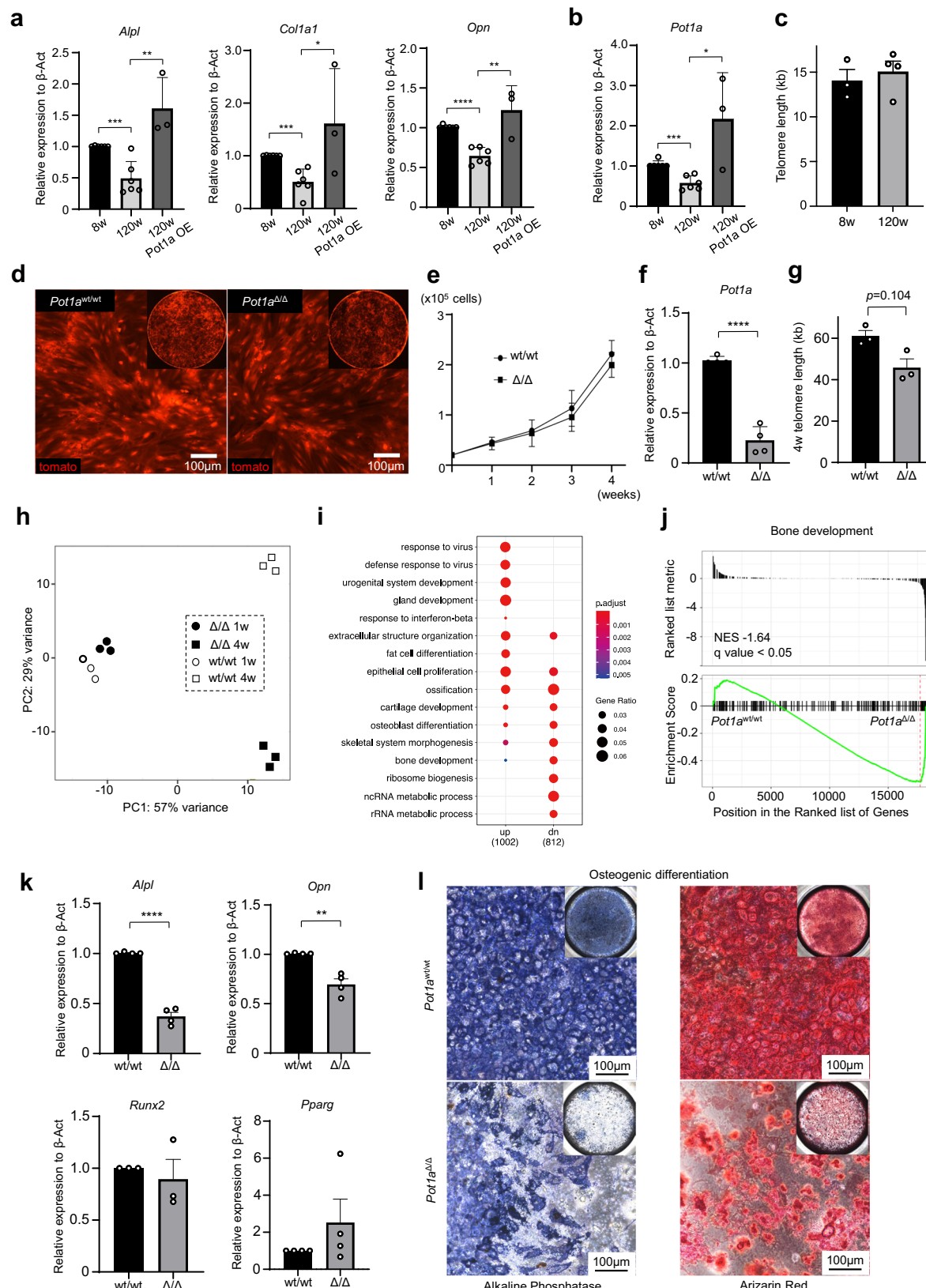

indicate that POT1a is essential for BM MSCs to maintain osteogenic differentiation potential.

**POT1a deficiency induces intracellular fatty acid accumulation in MSCs.** Previous studies reported that a peroxisome proliferator-activated receptor (PPAR) axis also has a critical role in lipid metabolism, promoting free fatty acid uptake and tria-cylglycerol accumulation in HSCs[20]. Therefore, we investigated metabolic pathways affected by POT1a deletion in MSCs cultured in vitro. Genes associated with the PPAR signalling pathway (including *Cd36*, whose gene product acts as a scavenger receptor

**Fig. 1 POT1a deletion in MSCs leads to impaired osteogenic-lineage differentiation. a, b** Expression of genes associated with osteogenic-differentiation (**a**) and POT1a (**b**) in BM MSCs sorted from 8–10 weeks and ~120 weeks old mice ($n = 6$ mice per group). 120w POT1a OE; MSCs from old mice were transfected with a retroviral vector to overexpress POT1a in culture ($n = 3$ mice per group). **c** Telomere length in BM MSCs sorted from 8–10 weeks old adult and ~120 weeks old aged mice ($n = 3$ mice per group). **d–g** Analyses of cultured MSCs isolated from Pot1a$^{w/w}$ or Pot1a$^{flox/flox}$ crossed with flox-stop-flox tdTomato mice. Representative images (**d**), cell proliferation (**e**; $n = 3$ mice per group), expression of POT1a (**f**; $n = 4$ mice per group) and telomere length (**g**; representative data of 3 independent experiments) after deletion of POT1a induced by administration of Cre recombinant vesicles in vitro. **h–j** RNA-sequencing analyses of MSCs, in which POT1a was deleted in vitro ($n = 3$ mice per group). **h** Principal component analysis of RNA-sequencing data, 1 and 4 weeks after deletion of POT1a in vitro. **i** Pathways significantly influenced 4 weeks after POT1a deletion, as assessed by GSEA. **j** Most of the genes included in the term "bone development" were down-regulated in POT1a deleted MSCs. **k** qPCR analyses of the genes related to osteogenic- and adipogenic- lineages in MSCs with POT1a deleted in vitro ($n = 3$ mice per group). **l** Representative images of osteocyte differentiation from MSCs with POT1a deleted in vitro. $*p < 0.05$, $**p < 0.01$, $***p < 0.001$.

essential for the uptake of fatty acids) were up-regulated 4 weeks after Cre administration (Fig. 2a, b and Supplementary Fig. 2a–c), suggesting that POT1a plays an important role in the lipid metabolism of MSCs.

Fatty acid oxidation (FAO) is a process that produces fatty acyl-CoA essential for mitochondrial oxidative phosphorylation (OXPHOS). Hence, we measured the oxygen consumption rate (OCR) in MSCs as an indicator of OXPHOS using an XF-24 extracellular flux analyser. The maximal respiration and spare respiratory capacities in POT1a-deleted MSCs were significantly lower than those of controls, while the amounts of intracellular mitochondria were comparable, when quantified using Mito-Tracker (Fig. 2c, d). The differences in the maximal and spare respiratory capacities were not prominent under treatment with Etomoxir, an inhibitor of fatty acid oxidation via negative control of Carnitine palmitoyl transferase 1a (Cpt1a), suggesting that FAO in mitochondria was inhibited upon POT1a deletion in MSCs (Fig. 2c). We also observed an accumulation of significantly more lipid droplets in Pot1a$^{\Delta/\Delta}$ MSCs compared to controls (as quantified by green fluorogenic BODIPY), which occurred presumably as a consequence of decreased FAO (Fig. 2e), although the expression of genes related to β-oxidation was not decreased (Supplementary Fig. 2c).

We previously reported that treatment with exogenous POT1a protein can inhibit excessive ROS in aged HSCs[7]. To trace the consequences of intracellular fatty acid accumulation, we analysed the production of reactive ROS and DDRs. ROS production in POT1a-deleted MSCs was significantly higher (Fig. 2f). Moreover, DDR, measured as the number of DNA foci that stained positive for 53BP1 (a protein that binds to chromatin at DNA double-strand break sites in the nuclei) was more prominently observed in the POT1a-deleted MSCs than controls (Fig. 2g). These results imply that POT1a is essential for mitochondrial energy production through the β oxidative pathway in BM MSCs and that accumulation of intracellular fatty acids by POT1a deficiency causes over-production of ROS. To investigate that increased production of ROS is responsible for impaired osteogenic-lineage differentiation, we performed the rescue experiments using an anti-oxidative agent, N-acetyl cysteine (NAC). Supplemented with NAC in culture media, the expression of genes related to osteogenesis, *Alpl* and *Opn*, was partially restored, suggesting ROS and consequent DNA damage at least in part contribute to the impaired osteoblast differentiation in POT1a deleted MSCs (Fig. 2h). These findings indicate that POT1a-deficient MSCs lose osteogenic differentiation potency through ROS production.

**POT1a-deficiency in NG2 positive cells causes impaired skeletal development**. To evaluate functions of POT1a in BM MSCs in vivo, we crossed Pot1a$^{fl/fl}$ mice with a NG2-cre strain (hereafter NG2-cre/Pot1a$^{fl/fl}$) that targets all pericytes in the whole body and

marks ~96% of Nestin-GFP positive MSCs in BM[14,16]. NG2-cre/Pot1$^{fl/fl}$ mice exhibited disproportional growth retardation—body sizes relative to brains were prominently smaller compared to littermates (Fig. 3a–d) and became moribund by 4 weeks after birth, suggesting severe skeletal retardation in the NG2-cre/Pot1$^{fl/fl}$ mice.

To further characterise skeletal integrity, we performed morphological measurements of the bones with micro computed tomography (micro CT) scans and microscopic imaging of bone sections. Micro CT analyses of femurs showed that development of the bones in metaphysis and trabeculae was impaired, although the membranous bones in epiphysis were still formed (Fig. 3e). Measured parameters, such as bone volume (BV) and bone mineral content (BMC), were significantly lower in NG2-cre/Pot1a$^{fl/fl}$ mice compared to control mice while no difference in the bone mineral density (BMD) was observed (Fig. 3f and Supplementary Fig. 3a). The numbers (Tb.N), thickness (Tb.Th) and volumes of the trabeculae (V*Tr) and BM (V*m.space) were significantly reduced in the femurs of NG2-cre/Pot1a$^{fl/fl}$ mice, while no difference in the trabecular separation (Tb.Sp) and trabecular spacing (Tb.Spac) was observed (Fig. 3g, h and Supplementary Fig. 3b).

We then evaluated bone morphology of the tibiae within sections stained with toluidine blue and found a significant decrease of cubic or conical osteoblasts along with thin osteoid in the NG2-cre/Pot1a$^{fl/fl}$ mice compared to control mice (Fig. 3i). Morphological analyses of the bone sections with toluidine blue staining showed that all of the parameters, osteoid surface per bone surface (OS/BS), osteoid volume per bone volume (OV/BV) and osteoblast surface per bone surface (Ob.S/BS), were significantly decreased in NG2-cre/Pot1a$^{fl/fl}$ mice, whereas eroded surface per bone volume (ES/BV), osteoclast surface per bone surface (Oc.S/BS) and osteoclast number per bone surface (N.Oc/BS) were not affected (Fig. 3j and Supplementary Fig. 3c), confirming the presence of impaired skeletal development. The proportion of putative PDGFRα$^+$ CD51$^+$ MSCs in the BM cells of NG2-cre/Pot1a$^{fl/fl}$ mice was increased, suggesting that presence of POT1a influences osteogenic- differentiation rather than maintenance of an MSC population (Fig. 3k and Supplementary Fig. 3d).

**BM microenvironments composed of POT1a-deleted MSCs impair B-cell development**. Since both bones and stroma cells, which are derived from MSCs, compose microenvironments for haematopoiesis in BM[21], we investigated whether the skeletal retardation observed in NG2-cre/Pot1a$^{fl/fl}$ mice also affected haematopoiesis. The total number of BM cells and the size of secondary hematopoietic organs, liver and spleen, were reduced in the NG2-cre/Pot1a$^{fl/fl}$ mice (Fig. 3k and Supplementary Fig. 4a–c). To examine haematopoiesis further we used single-cell RNA sequencing (DropSeq)[22,23]. Cluster analysis of the gene expression profiles obtained from whole BM cells revealed the

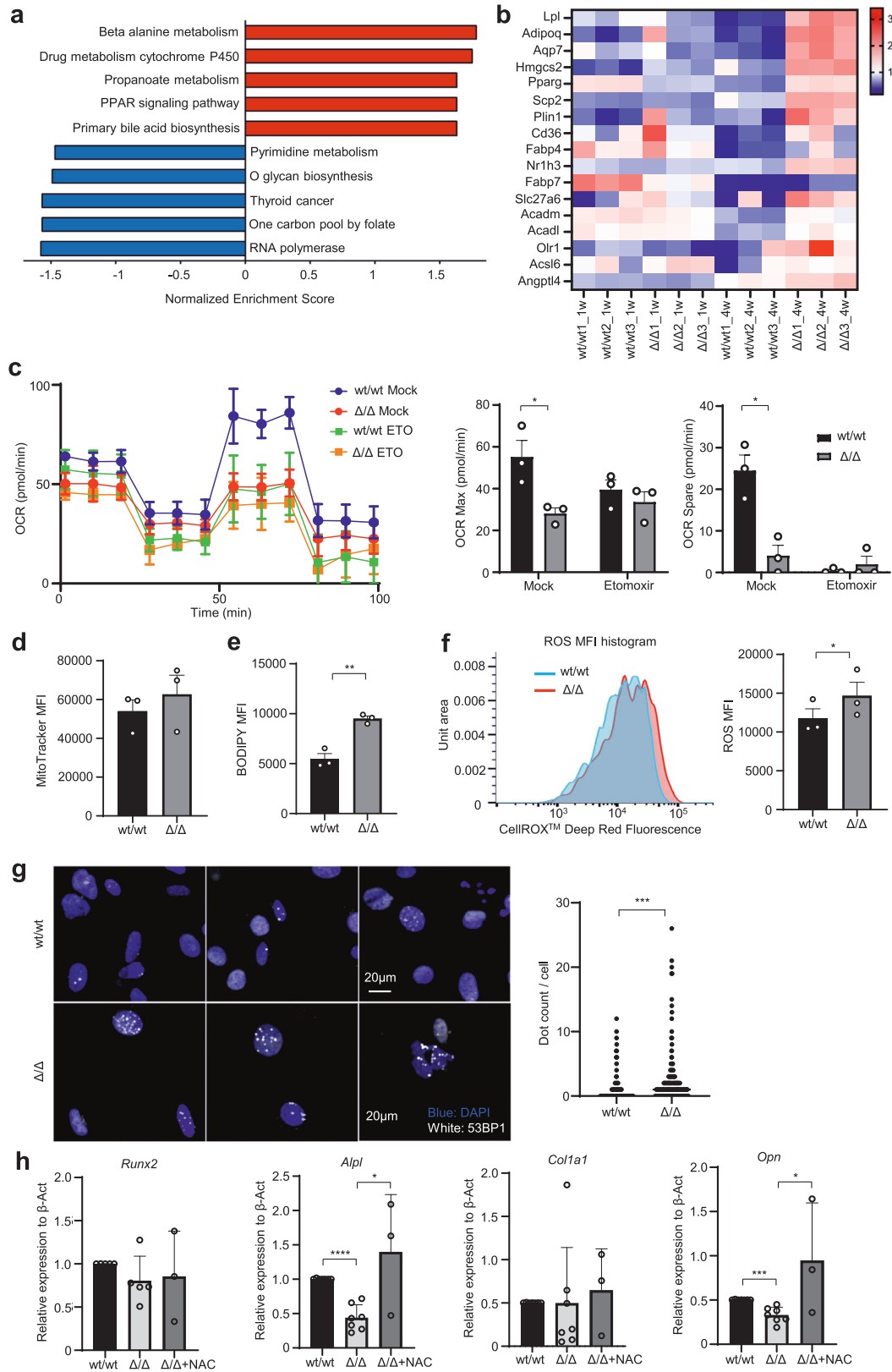

presence of 15 cell subgroups (Fig. 4a). We compared the relative sizes of these subpopulations between the NG2-cre/Pot1a^fl/fl and control mice and found a significant reduction in cells of the B-cell lineage in NG2-cre/Pot1a^fl/fl marrow (Fig. 4a, b). Cell subpopulations associated with B-cell differentiation from hematopoietic stem/progenitor cells (HSPCs) were identified

according to the expression of established marker genes (Fig. 4c and Supplementary Fig. 4d). The reduction in relative cell numbers in the NG2-cre/Pot1a^fl/fl mice was observed evenly across all stages of B-cell development, suggesting that B-cell differentiation was affected at from immature stages onwards (Fig. 4c).

**Fig. 2 POT1a deficiency induces intracellular fatty acid accumulation in MSCs. a, b** Gene expression differences from RNA-sequencing analysis one and 4 weeks after deletion of POT1a in vitro. **a** Pathways related to cell metabolism from the KEGG database. **b** Heatmaps of expression of genes associated with the peroxisome proliferator-activated receptor (PPAR) pathway in the KEGG database 1 and 4 weeks after deletion of POT1a. **c** Measurements of the OCR in POT1a deleted MSCs, cultured with or without Etomoxir by XF-24 extracellular flux analyser. Real-time OCRs were determined during sequential treatments with oligomycin (an ATP synthase inhibitor), FCCP, and antimycin-A/rotenone. Representative data from three independent experiments is shown. **d** FACS analysis of amounts of mitochondria detected with MitoTracker™ DeepRed. Representative data of three independent experiments is shown. **e** Quantification by FACS of intracellular fatty acid accumulation labelled with green fluorogenic BODIPY. Representative data of three independent experiments is shown. **f** Measurements of reactive oxygen species (ROS) in POT1a deleted MSCs by flow cytometry. ROS was detected with CellROX. Representative histogram and quantification of the mean fluorescence intensity is shown. **g** Immunocytochemical staining of DDR measured as foci stained with 53BP1 in MSCs with POT1a deleted in vitro. Three hundred of cells per group were analysed and the numbers of the foci were plotted. **h** Expression of genes associated with osteogenic differentiation in MSCs with POT1a deleted in vitro cultured supplemented with or without NAC ($n = 3$ mice per group). *$p < 0.05$, **$p < 0.01$, ***$p < 0.001$.

To further determine which stages of hematopoietic differentiation were influenced, we independently analysed the frequency of HSPCs in the BM of NG2-cre/Pot1a$^{fl/fl}$ mice by flow cytometry (Supplementary Fig. 4e, f). The frequencies of CD150$^+$CD48$^-$ HSCs were comparable to controls and the preservation of HSC activity was confirmed with competitive repopulation assays (Supplementary Fig. 4e, g, h). We also examined the frequencies of more committed progenitors including a common lymphoid progenitor (CLP) and found no significant differences between the NG2-cre/Pot1a$^{fl/fl}$ and control marrow (Supplementary Fig. 4f). We then examined series of progenitors over the course of B-cell development in the BM by flow cytometry and found that B-lymphopoiesis was predominantly impaired at the early stage after pre/pro B cells emerged (Fig. 4d–f and Supplementary Fig. 4i). In the peripheral blood, counts of white blood cells (WBC), especially B lymphocytes rather than T cells or myeloid cells, significantly decreased whereas no significant difference was observed in haemoglobin and platelet levels in the NG2-cre/Pot1a$^{fl/fl}$ mice, confirming the presence of selective defects of B lymphopoiesis (Fig. 4g–i and Supplementary Fig. 4j). To confirm these results, we performed CFU assays. The BM cells from the NG2-cre/Pot1a$^{fl/fl}$ mice gave rise to smaller sizes of CFU-PreB colonies and their numbers were significantly reduced compared to the controls whereas CFU-GM colonies were comparably observed. (Fig. 4j and Supplementary Fig. 4k).

**POT1a deficiency in MSCs impacts bone growth.** To further trace the POT1a deleted MSCs in vivo, we generated an NG2-cre/loxp-tdTomato/Pot1a$^{fl/fl}$ mouse line. The tdTomato signals were observed exclusively in the CD45$^-$TER119$^-$CD31$^-$PDGFRα$^+$CD51$^+$ subpopulation of the whole BM (Fig. 5a and Supplementary Fig. 5a). The frequencies of tdTomato$^+$ cells were no different between the NG2-cre/loxp-tdTomato/Pot1a$^{fl/fl}$ and control groups although their absolute numbers were significantly lower than in controls (Fig. 5b, c). Through image analysis of the bone and BM from the NG2-cre/loxp-tdTomato/Pot1a$^{fl/fl}$ mice, tdTomato$^+$ cells were found to be distributed in the growth plates, endosteum and throughout the BM space (Fig. 5d). The alignment of the growth plates and the bone lining cells appeared to be disturbed, indicating that the findings observed in the bone sections (Fig. 3i) are likely attributed to defects in differentiation potential of POT1a-deficient MSCs.

To characterise POT1a-deleted MSC-derived cells in vivo, we sorted the tdTomato$^+$ cells in the BM digested from NG2-cre/loxp-tdTomato/Pot1a$^{fl/fl}$ mice, at the age of 2 weeks and performed RNA-sequencing analyses. The expression of POT1a was reduced in tdTomato$^+$ MSCs (Fig. 5e) and GSEA confirmed that several pathways related to bone differentiation and development were down-regulated in tdTomato$^+$ cells (Fig. 5f). Together, these results indicate that osteo-differentiation in BM

MSCs is suppressed by POT1a deficiency that leads to impaired skeletal development in vivo.

To elucidate mechanisms of the impaired B-lymphopoiesis in the BM, we investigated expression of genes related to B-cell development in the BM MSCs and bone lining osteoblasts (BLCs) that are reported to support B lymphopoiesis by producing important factors such as IL-7[24,25]. We sorted MSCs and BLCs from the digested BM and bones of NG2-cre/loxp-tdTomato/Pot1a$^{fl/fl}$ mice (Fig. 5g) and measured IL-7 expression by qPCR. Expression of IL-7 in the MSCs and BLCs was not significantly affected in NG2-cre/loxp-tdTomato/Pot1a$^{fl/fl}$ mice (Fig. 5h).

In summary, these results suggest that BM microenvironments composed of POT1a deficient MSCs are unable to support B-lymphopoiesis due to reduced number of bone lining osteoblasts.

## Discussion

In this study, we have demonstrated the importance of POT1a in maintaining differentiation potential of MSCs towards the osteogenic-lineages. POT1a deficient MSCs exhibited decreased mitochondrial OXPHOS and intracellular fatty acid accumulation. Previous studies have revealed the association between telomere dysfunction and impaired mitochondrial biogenesis[26]. Telomere dysfunction induces a profound p53-dependent repression of the master regulators of mitochondrial biogenesis, and peroxisome proliferator-activated receptor gamma coactivators (PGC)-1α and PGC-1β, leading to bioenergetic compromise due to impaired OXPHOS and ATP generation[26,27]. Since osteoblast differentiation requires upregulation of mitochondrial OXPHOS in MSCs in order to supply ATP through increased β-catenin acetylation[28], decreased mitochondrial OXPHOS may contribute in part to the impaired bone development observed in NG2-cre/Pot1a$^{fl/fl}$ mice.

RNA-sequencing analysis revealed that genes associated with the PPAR pathway (including *Cd36*) were up-regulated in POT1a deficient MSCs. Because intracellular uptake of fatty acids is a process that produces fatty acyl-CoA essential for mitochondrial OXPHOS, these results suggest that enhanced PPAR pathway activity under impaired mitochondrial OXPHOS might result in intracellular accumulation of fatty acids and, consequently, excessive ROS production.

Notably, the series of cellular events triggered by POT1a deficiency in MSCs may recapitulate the physiological decline of stem cell function with ageing[29]. POT1a deficiency in MSCs appears to affect differentiation potential rather than cell proliferation, arguing that cell differentiation requires additional metabolic activation while steady cell growth can be achieved with baseline oxygen consumption as reported with HSC differentiation[30,31]. In our experiments, there was no significant effect on POT1a deficient MSCs cellular growth in vitro despite mitochondrial

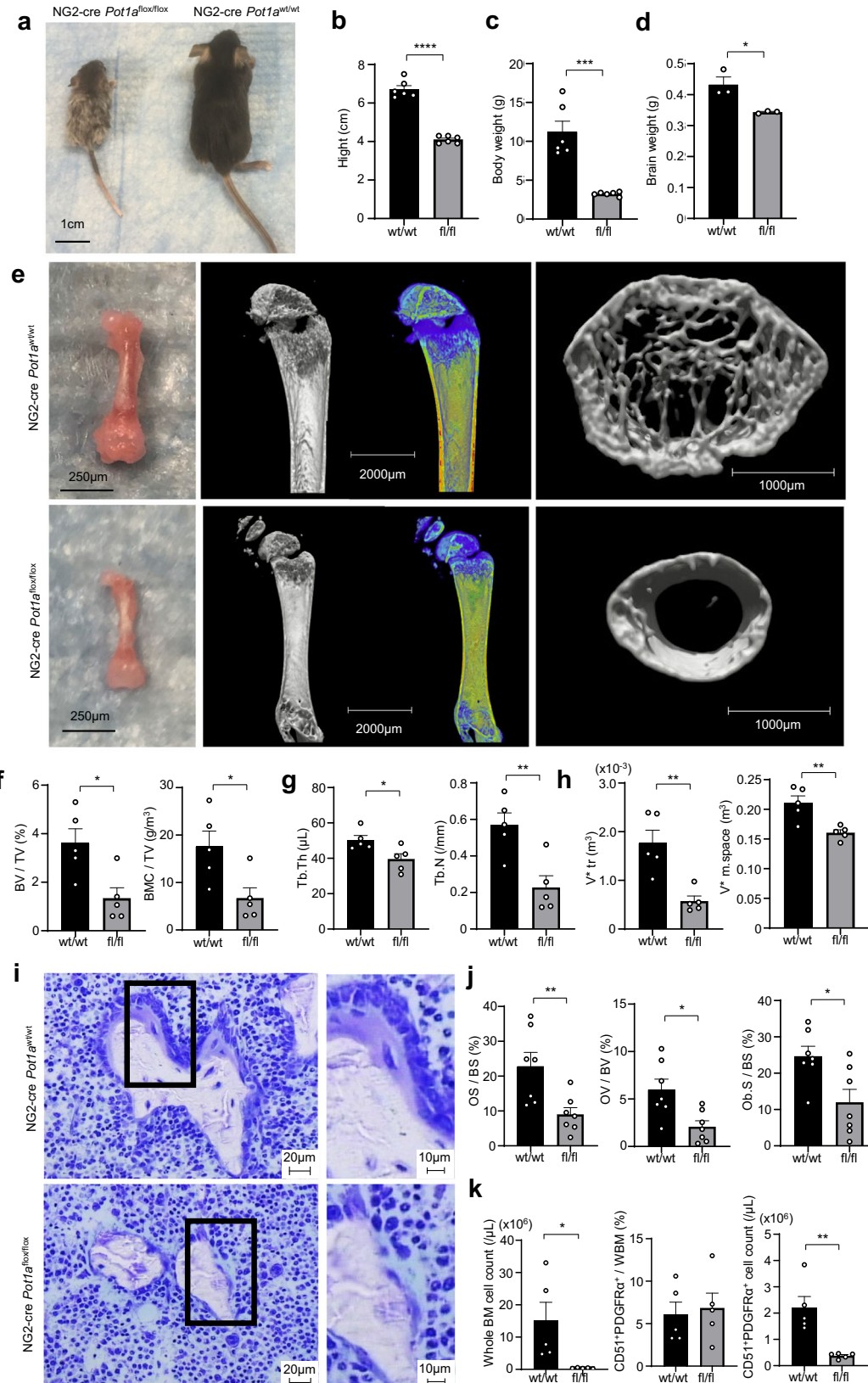

dysfunction and decreased FAO. It is presumed that nucleotide synthesis is compensated by alternative pathways we have not identified in this study.

Previous studies have reported that telomere dysfunction and impaired mitochondrial function are associated with an ageing-related functional decline of the heart[27,32]. Accumulation of fatty acids is also reported to provoke myocardial damage and heart failure in metabolic disorders such as obesity and diabetes[27]. Among the genes involved in the PPAR pathway, PPARγ, which belongs to the nuclear hormone receptor superfamily of ligand-activated transcription factors, is also a key transcription factor for adipogenic differentiation[33]. Previous studies have reported

**Fig. 3 MSC-targeted POT1a deficient mice exhibit impaired skeletal development. a** Representative appearance of NG2-cre/Pot1a[fl/fl] mice.
**b**–**d** Quantification of heights (**b**), body weights (**c**) and brain weights (**d**) of the NG2-cre/Pot1a[fl/fl] mice compared to littermate controls. ($n = 6$ mice per group). **e** Micro CT analyses of bones. **f**–**h** Comparisons of the measured parameters: bone volume (BV)/tissue volume (TV); bone mineral content (BMC)/TV; thickness (Tb.Th); number (Tb.N); volume of the trabeculae (V*Tr) and BM (V* m.space) ($n = 5$ mice per group). **i** Morphological measurements of bone sections stained with toluidine blue. **j** Comparisons of the measured parameters: osteoid surface/bone surface (OS/BS); osteoid volume/bone volume (OV/BV) and osteoblast surface/bone surface (Ob.S/BS) ($n = 7$ mice per group). **k** Quantification of numbers per femur of whole BM cells and frequencies and numbers per femur of PDGFRα[+]CD51[+] MSCs in the BM of NG2-cre/Pot1a[fl/fl] mice ($n = 3$ mice per group). $*p < 0.05$, $**p < 0.01$, $***p < 0.001$, $****p < 0.0001$.

that overexpression of PPARγ2 in cultured fibroblasts promotes adipogenic differentiation[34] and that a subset of osteoblasts with high PPARγ expression maintain adipocyte differentiation potential[35]. In addition, osteogenic inducers such as bone morphogenetic protein (BMP) and Wnt are known to inhibit PPARγ transactivation during osteogenic-lineage differentiation, while up-regulation of PPARγ inhibits terminal differentiation in the dexamethasone-induced osteogenic differentiation model[36,37]. This competition between the osteogenic and adipogenic differentiation supports the notion that impaired osteogenic- differentiation observed in the POT1a deficient MSCs is attributed, in part, to upregulation of PPARγ.

Genotoxic stresses such as impaired DNA repair and excessive ROS due to telomere attritions induce senescence, apoptosis and mitochondrial dysfunction in stem cells through activation of p53[38]. In HSCs, inactivation of TPP1, another component of the shelterin complex, upregulates transcription of p53-targeting genes and induces cell cycle arrest[39]. A homolog of the mouse POT1a gene, POT1b, also plays a role in protecting the C-strand of telomeres from extensive nucleolytic degradation. Dysfunctional telomeres caused by loss of POT1b in HSCs initiate a p53-dependent DDR and apoptotic response that ultimately results in a progressive decrease in the number of HSCs[40]. In light of these results, our finding that POT1a-deficient MSCs lose their multilineage differentiation capacity highlights the importance of telomere protection for their maintenance of osteogenic differentiation potential.

In BM, HSC niches are hosted by stromal cells exhibiting MSC activity[13,14,41–43] whereas osteoblasts, a progeny of MSCs, are reported to host lymphoid committed progenitors[21]. Several studies report that mature osteoblasts have important roles in supporting B lymphopoiesis at the stages of early transition from HSPCs and late maturation of common lymphoid progenitor to IgM[+] B cells[44,45]. Mechanistically, osteoblast-derived IL-7 supports the late stage of B cell (but not T cell) development in the BM[25]. Recent studies have revealed that systemic inflammation, for instance, caused by sepsis, reduces the number of IL7-producing osteoblasts, which, in turn, diminishes the production of common lymphoid progenitors from HSCs[24]. Here, we found that the number of bone lining osteoblasts were also reduced in NG2-cre/Pot1a[fl/fl] mice, suggesting that IL-7 may be insufficient to support B cell development.

Our findings are also of clinical significance, since NG2-cre/Pot1a[fl/fl] mice exhibit a similar phenotype to dyskeratosis congenita as Tin2 deficient animals, including comparable skeletal defects and hematopoietic failures[46]. A previous report has shown that human BM skeletal stem/progenitor cell defects contribute to BM failure in telomere biology disorders[47]. Telomere dysfunction in BM MSCs may similarly contribute to the pathology of hereditary BM failure syndrome. Our previous study shows that treatment with exogenous POT1a protein can rejuvenate aged mouse HSCs[7]. Therefore, introducing factors essential for telomere protection, such as POT1, may be an important potential therapeutic option for human patients with telomere biology disorders.

## Methods

**Mice.** C57BL/6 (B6-Ly5.2) were purchased from CLEA Japan, Inc. Congenic C57BL/6 mice for the Ly5 locus (B6-Ly5.1) were purchased from Sankyo lab. POT1a[fl/fl] (B6;129-Pot1a[tm1.1Tdl]/J), NG2-cre (B6.FVB-Tg(Cspg4-cre)1Rkl/J) and flox-stop-flox tdTomato (B6.Cg-Gt(ROSA)26Sor[tm9(CAG-tdTomato)Hze]/J) mice were provided by the Jackson laboratory. The Gene recombination experiment safety committee and animal experiment committee in Kyushu University approved this study. All experiments were carried out in accordance with ARRIVE Guideline and the Guidelines for Animal and Recombinant DNA experiments at Kyushu University.

**Mouse antibodies.** The following monoclonal antibodies (Abs) were used for flow cytometry and cell sorting: anti-c-Kit (2B8, BD Biosciences), -Sca-1 (E13-161.7/D7, BioLegend), -CD140a (APA5, BD Biosciences), -CD51 (RMV-7, BIO-RAD), -CD4 (RM4-5, BioLegend), -CD8a (53-6.7, BioLegend), -B220 (RA3-6B2, BioLegend/WAKO), -CD31 (390, BioLegend), -TER-119 (TER-119, BioLegend/BD Biosciences), -Gr-1 (RB6-8C5, BD Biosciences), -Mac-1 (M1/70, BioLegend), -CD48 (HM48-1, BD Biosciences), -CD150 (TC15-12F12.2, BioLegend), -CD19 (6D5, BioLegend), -IgM (MHM-88, BioLegend), -CD43 (S11, BioLegend), -CD45.1 (A20, BD Biosciences), -CD45.2 (104, BD Biosciences), -CD45 (30-F11, BioLegend), -Flt3 (A2F10.1, BD Biosciences), -IL7Rα (A7R34, TONBO biosciences), -FcγR (2.4G2, TONBO biosciences), -53BP1 (Novus Biologicals). A biotinylated CD34 (RAM34, Thermo Fisher Scientific) and a fluorochrome labelled streptavidin (BD Biosciences/BioLegend) were used.

**Blood collection and complete blood count analyses.** Mice were sacrificed and blood was collected by cardiac puncture. CBC analyses were performed using an automatic blood cell analyser K-4500 (Sysmex).

**Cell preparation and flow cytometry.** BM cells were isolated from femurs and tibiae by flushing with PBS without $Ca^{2+}$ and without $Mg^{2+}$ (PBS(−)). Peripheral blood was collected from tail vein. Stained cells were analysed and sorted using a FACS AriaIII (BD Biosciences). Data were analysed with FlowJo software (BD Biosciences).

**Colony forming assay.** $0.5 \times 10^5$ or $1.5 \times 10^5$ BM cells were plated per well in the media, MethoCult™ GF M3534 for CFU-GM, and MethoCult™ M3630 for CFU-PreB, respectively. Forming colonies were counted after 7–10 days culture in a 37 °C, 5% $CO_2$ incubator.

**Isolation of BM MSCs.** BM cells were flushed out from femurs and tibiae with HBSS solution made by dissolving collagenase IV (0.02 g/mouse, Gibco) and Dispase (0.04 g/mouse, Gibco) and digested for 20 min at 37 °C. After haemolysis using ammonium chloride, CD45[−]TER119[−]CD31[−]CD51[+]PDGFRα[+] cells were

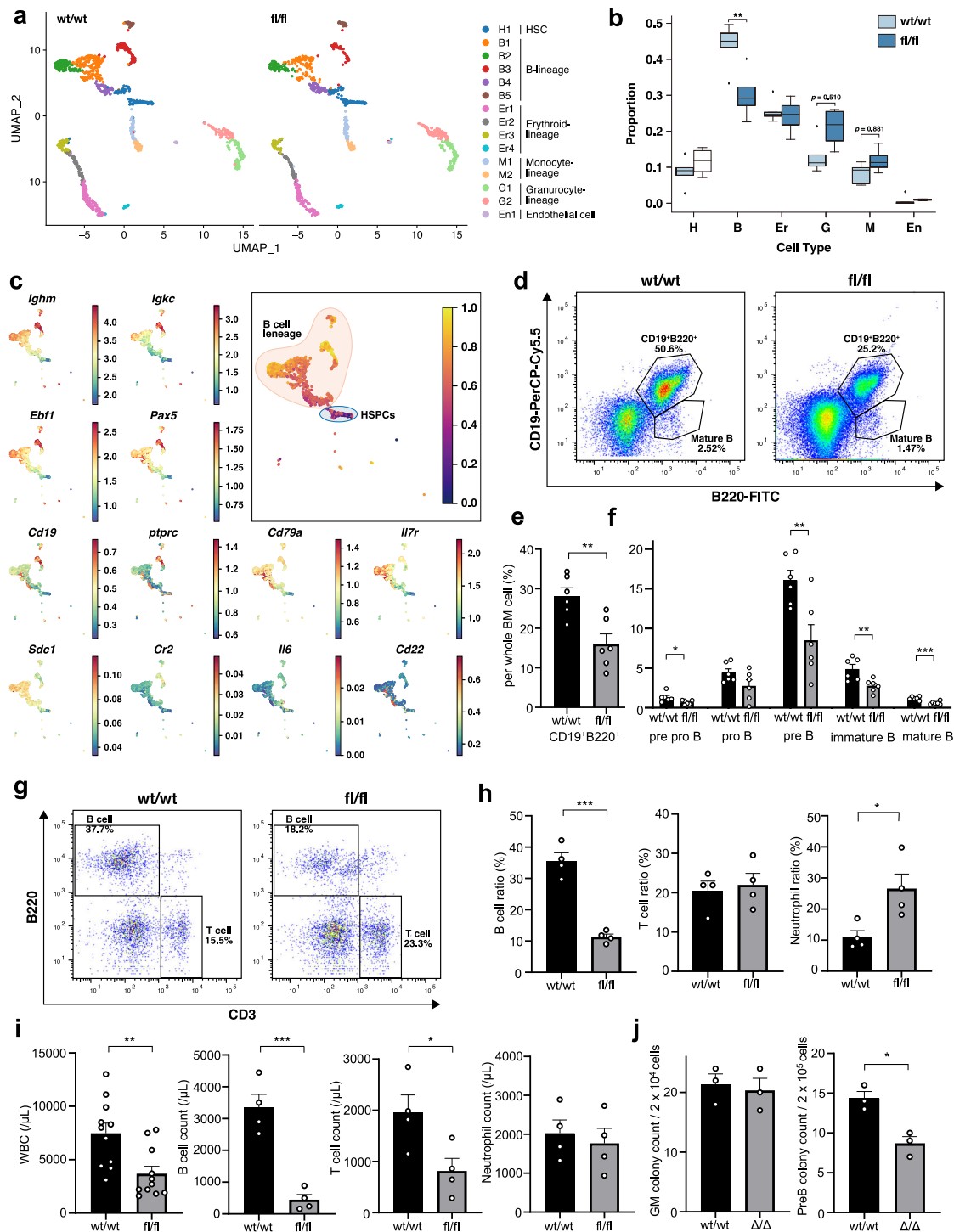

**Fig. 4 BM microenvironments composed of POT1a-deleted MSCs impair B-lymphopoiesis. a, b** Two-dimensional representation (UMAP) of single-cell RNA sequencing data (DropSeq) from unselected BM cells in NG2-cre/Pot1a$^{fl/fl}$ or wildtype mice. Cells were divided into 15 clusters using Louvain clustering, based on individual gene expression profiles ($n = 3$ mice per group). **c** Pseudotime analysis for B-cell differentiation (top right) and expression heatmaps of genes related to the B-cell lineage (left and bottom right). **d–f** FACS analysis of B-cell committed progenitors in the NG2-cre/Pot1a$^{fl/fl}$ marrow. Representative FACS plots (**d**), and quantification **e**, **f** ($n = 4$ mice per group). **g–j** Analysis of the peripheral blood from NG2-cre/Pot1a$^{fl/fl}$ mice. **g, h** FACS analysis of the subpopulations in the WBCs. Representative FACS plots and quantification of their frequencies ($n = 4$ mice per group) area shown. **i** Absolute counts of WBCs ($n = 11$ mice per group) and subpopulations measured by FACS ($n = 4$ mice per group). **j** Numbers of colonies formed from the NG2-cre/Pot1a$^{fl/fl}$ marrow cells cultured in the media for CFU-GM, and CFU-PreB, respectively ($n = 3$ mice per group). *$p < 0.05$, **$p < 0.01$, ***$p < 0.001$.

used as MSCs. For some experiments, MSCs were enriched with a MACS Separator using CD45/Ter119 MicroBeads (Miltenyi Biotec). The flow-through fraction was used as an MSC-enriched population.

**In vitro culture of MSCs.** BM cells flushed out from mouse femurs and tibiae were cultured on 6-well-plate for 1 week in MesenCult expansion kit media (STEMCELL technologies). Adherent cells were collected by cell detachment solution

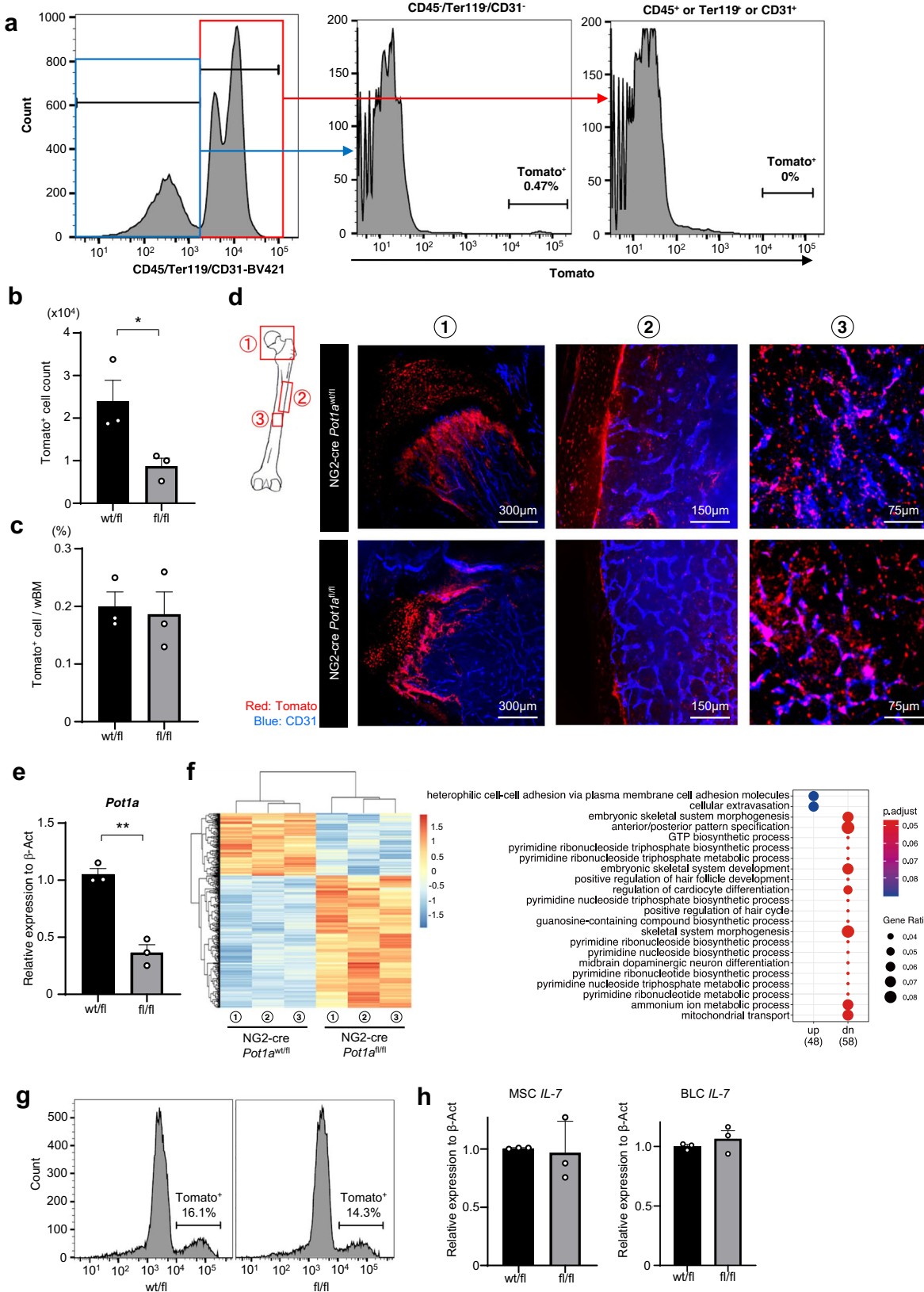

Accutase (nacalai tesque) on day 7, CD51+/PDGFRα+ cells were sorted and further cultured on 24-well-plates treated by the vacuum gas plasma for 3 days. Cre Recombinase Gesicles (CLN) were applied at 10 μL/well and exposed for 12 h. After 48 h of culture, tdTomato fluorescently coloured cells were re-sorted. The sorted cells were passaged once per week at 100% confluency and cultured in a 37 °C, 5% $O_2$ and 10% $CO_2$ incubator for about 4 weeks and used for gene analyses or differentiation experiments. For POT1a overexpression, the Plat-E packaging cell line was transfected with the previously described POT1a overexpression vector[7] using FuGene HD (Promega). The harvested culture supernatant was concentrated in 10 folds with PEG-it (System

**Fig. 5 POT1a deficiency in BM MSCs has impacts on bone growth.** Analyses of the BM from NG2-cre/loxp-tdTomato/Pot1a$^{fl/fl}$ mice. **a** FACS plots of tdTomato positive cells in the CD45$^-$TER119$^-$CD31$^-$ population from whole BM. **b**, **c** Numbers and frequencies of tdTomato$^+$ cells in whole BM ($n = 3$ mice per group). **d** Whole-mount images of epiphysis (1), diaphysis (2) and marrow cavity (3) of the femur. Endothelial cells are stained with anti-CD31 antibody i.v. **e** qPCR analysis of POT1a expression in tdTomato$^+$ cells in the BM ($n = 4$ mice per group). **f** RNA-sequencing analysis of sorted tdTomato$^+$ cells from the BM of NG2-cre/loxp-tdTomato/Pot1a$^{fl/fl}$ mice. The heat map shows differentially expressed genes and significantly enriched pathways by GSEA ($n = 3$ mice per group). **g**, **h** qPCR analyses of IL-7 related to B-cell development in tdTomato$^+$ cells sorted from the BM of NG2-cre/loxp-tdTomato/Pot1a$^{fl/fl}$ mice. The tdTomato$^+$ cells isolated from the BM cells and the bones were analysed separately as MSCs and bone lining cells (BLCs, namely ostaoblasts) ($n = 3$ mice per group). *$p < 0.05$, **$p < 0.01$.

Biosciences, LLC) and used for further experiments. The sorted CD51$^+$/PDGFRα$^+$ cells were infected by adding the supernatant containing and cultured for 4 weeks for the analyses. For NAC rescue, NAC was added to media in 10 μg/ml of concentration and the MSCs were cultured in the media for 4 weeks before the analyses.

**In vitro MSC differentiation assay.** To evaluate tri-lineage differentiation in vitro, MSCs were cultured by Mouse Mesenchymal Stem Cell Functional Identification Kit (R&D SYSTEMS). For osteocyte differentiation, MSCs were seeded at a density of $3 \times 10^3$ cells at 96-well-plate and incubated in a 37 °C, 5% $CO_2$ incubator. At 50–70% confluency, the media in each well was replaced with 200 μl of Osteogenic Differentiation Media to induce osteogenesis. The medium was changed every 3–4 days. After 21 days, the cells were fixed and stained with Alizarin Red. For adipocyte differentiation in vitro, the media in each well was replaced with 200 μl of MEM alpha Basal Media to induce adipogenesis. The medium was changed every 2–3 days. After 14 days, the cells were fixed and stained with Oil Red. For chondrocyte differentiation, MSCs are transferred at a density of 250,000 cells to a 15 ml conical tube. The transferred cells were centrifuged at 200 g for 5 min at room temperature and incubated at 37 °C and 5% $CO_2$ in 0.5 ml of Chondrogenic Differentiation Media. The medium was changed every 3–4 days. After 17–21 days, a chondrogenic pellet was immersed in 70% ethanol, fixed at 4 °C for 1 week. Samples were stained with H&E and Alcian Blue by Applied Medical Research Co., Ltd. (Osaka, Japan).

**Q-PCR.** Q-PCR was performed with CFX384 Touch real-time PCR analysis system (Bio-Rad) using TaqMan Fast Universal PCR master mixture (Thermo Fisher Scientific). The TaqMan® Gene Expression Assay mixes used in this study are listed in Supplementary Table. β-actin was used as an internal control. Data were analysed by CFX Manager Software v2.1. All experiments were carried out in quadruplicate.

**Quantitative telomeric repeat amplification.** Telomere length was measured with Absolute Human Telomere Length Quantification qPCR Assay Kit (ScienCell). In this assay, sorted cells were lysed, and the lysate was added to telomeric repeats onto a substrate oligonucleotide. The resultant extended product was subsequently amplified by PCR. Direct detection of PCR product was monitored on CFX384 Touch real-time PCR analysis system.

**OCR.** BM-MSCs were analysed using an XF-24 Extracellular Flux Analyzer (Seahorse Bioscience). Briefly, BM-MSCs were seeded in XF-24 well culture plates (10,000 cells/well). At the specified time points, BM-MSCs were washed and analysed in XF Running Buffer (unbuffered DMEM medium with 10 mM glucose, 10% foetal calf serum, and 2 mM L-glutamine) according to the manufacturer's instructions to obtain real-time measurements of the OCR. Analyses of the OCR in response to 0.25 μM

oligomycin, 10 μM FCCP, and 1 μM rotenone plus 1 μM antimycin A were performed.

**Library preparation for RNA sequencing.** mRNA was isolated from sorted tdTomato$^+$ MSCs with NucleoSpin® RNA XS (Takara Bio USA, Inc.) and cDNA was synthesized using SMART-Seq® v4 Ultra® Low Input RNA Kit for Sequencing (Takara Bio USA, Inc.). The full-length cDNA was processed with the Nextera XT DNA Library Preparation Kits (Illumina). The amplified cDNA was validated using the Agilent 2100 Bioanalyzer and Agilent's High Sensitivity DNA Kit (Agilent).

**Library preparation for single-cell RNA sequencing.** After pre-run setup, the pump of the cells, then the beads, and finally the oil were started. After assessment of droplet quality and bead droplets, the beads were in the whiter layer on top. Through the breakage treatment, the white beads were gathered and washed with 5X RT buffer. cDNA strands were generated on the RNA hybridized to the bead primer with reverse transcription mix. After exonuclease I treatment by incubating at 37 °C for 45 min with rotation, PCR program was exceeded by 95 °C 3 min—4 cycles of 98 °C 20 s, 65 °C 45 s, 72 °C 3 min—9 cycles of 98 °C 20 s, 67 °C 20 s, 72 °C 3 min—72 °C 5 min; 4 °C hold. The purified cDNA by AMPure XP beads was measured by Bioanalyzer. Single-cell sequencing was performed with some modifications. The hydrophobic surface treatment of polydimethylsiloxane (PDMS) microfluidic devices was administered by incubating channels with 1% Trichloro (1H,1H,2H,2H-perfluorooctyl) silane (Sigma-Aldrich, 448931) in Fluorinert FC-40 (Sigma-Aldrich, F9755) for 5–10 min at RT. The sequencing library was prepared with pre-amplification using $4 + 12$ PCR cycles (95 °C 3 min—4 cycles of: 98 °C 20 s; 65 °C 45 s; 72 °C 3 min—12 cycles of: 98 °C 20 s; 67 °C 20 s; 72 °C 3 min—72 °C 5 min; 4 °C hold). Pooled libraries were sequenced on the NextSeq 500 system using a 75-cycle kit (read 1, 20 cycles; index 1, eight cycles; read 2, 50 cycles).

**Micro CT and bone morphometry.** Femurs and tibiae of 3-week-old mice were harvested and immersed in 70% ethanol and fixed at 4 °C for 1 week. Anatomical analysis and measurements of the femurs and tibiae using micro CT and bone morphology measurement by toluidine blue staining were performed by Isozo Co., Ltd. (Tokyo, Japan). Micro CT scanning was performed with a ScanXmate-D090S105 (Comscantechno). Three-dimensional microstructural image data were reconstructed and structural indices were calculated using a TRI/3D-BON software (Ratoc System Engineering). For the micro CT measurements, areas with the width of 1.0 mm from 100 μm away from the lowermost end of the epiphysis were set. For the analysis of osteoblasts, osteocytes and osteoclasts, the undecalcified tibiae were embedded in glycol methacrylate, sectioned and stained with toluidine blue and tartrate-resistant acid phosphatase (TRAP), respectively. The parameters for osteoblasts, osteocytes and osteoclasts in the secondary trabecular bone of tibiae were assessed in microscopic fields from two sections per mouse by moving the sections in

equally sized steps along the x and y axes. Images were acquired by a light microscope (Axio Imager 2; Zeiss) and all histological analyses performed using a WinROOF 2013 software (Mitani).

**Intracellular labelling for flow cytometry**. For intracellular staining, cells were first stained with cell surface markers prior to fixation and permeabilization. Cells were then fixed and permeabilized with Cytofix/Cytoperm ™ (BD Biosciences). ROS level was measured by CellROX™ Deep Red Reagent (Thermo Fisher Scientific). Intracellular mitochondria were measured by Mito-Tracker™ Deep Red (Thermo Fisher Scientific). For measuring Intracellular fat accumulation by BODIPY™ (Thermo Fisher Scientific), BM-MSCs were fixed with 4% paraformaldehyde (Wako Pure Chemical Industries) and permeabilized with saponin (0.2% saponin and 0.5% BSA in PBS) for 30 min at room temperature. BM-MSCs were incubated at room temperature with 50 µg/mL BODIPY™. Flow cytometry analysis was performed with a FACS Verse using FACSuite software (BD Biosciences).

**Immunocytofluorescence imaging**. Cells were spread onto glass slides and fixed in methanol. After blocking with Protein Block (DAKO), samples were incubated at 4 °C overnight with anti-53BP1 (Novus Biologicals) as primary Abs After washed with PBS(−), samples were stained with AF647 Donkey anti-Rabbit antibody for 2 h. For nuclear staining, specimens were treated with DAPI (Molecular Probes). Fluorescence images were acquired using a ZEISS AXIO examiner D1 microscope (ZEISS) equipped with a LAZER spinning disc confocal head (CSU-W1, Yokogawa). The imaging data were analysed with Image-Pro software (Media Cybernetics).

**Immunohistofluorescence imaging**. For in vivo staining of BM endothelial cells, anti-CD31 and -CD144 Alexa Fluor 647-conjugated Abs (Alexa Fluor 647) were intravenously injected. Mice were perfused with PBS(−) for 10 min after injection to remove unbound reagents and long bones were collected for immunofluorescence imaging. For preparations of frozen sections of long bones, femoral or tibial bones were perfusion-fixed and then post-fixed for overnight in 4% PFA, embedded in an optical cutting temperature (OCT) compound and freezed at −100 °C. The frozen tissues were shaved on a cryostat (LEICA CM3050 S) until the BM cavity was fully exposed. The tissues were carefully harvested from melting an OCT compound. The images were acquired using a ZEISS AXIO examiner D1 microscope (ZEISS) equipped with a LAZER spinning disc confocal head (CSU-W1, Yokogawa). The imaging data were analysed with Image-Pro software (Media Cybernetics).

**Long term reconstitution assay**. Donor Ly5.1$^+$ BM mono nuclear cells (MNCs) ($2 \times 10^6$ cells per mice) were transplanted into lethally irradiated (total 9.5 Gy) Ly5.2 mice using $2 \times 10^6$ competitor Ly5.2$^+$ BM MNCs. The percentages of donor-derived cells (Ly5.1$^+$) in peripheral blood were analysed monthly by flow cytometry. 16 weeks after BM transplantation, chimerism of donor-derived B, T, and myeloid cells in peripheral blood was analysed.

**Collagenase release of bone lining cells from mice BM**. After BM cells were flushed out with PBS(-), trabecular bone fragments were incubated in Collagenase type 1 (Worthington Biochemical, LS004196) solution (collagenase type 1: 37.5 mg, 20% DMEM: 15 ml, and 10 units/µL DNase: 37.5 µL per 2 mice) for total 90 min at 37 °C under continuous rotation. Bone fragments were washed with wash solution (PBS: 50 ml, FBS: 1 ml, 1 M HEPES:

10 µL). After haemolysis, cells released from the bones were sorted using a FACS AriaIII (BD Biosciences).

**RNA sequencing data analysis**. For each sample, the library was sequenced on the NextSeq 500 system using 75-cycle kit. The reads were mapped against the mouse reference genome (mm10) by HISAT2 (version 2.0.4)[48], and the expression levels of all genes were quantified by using featureCounts (version 1.6.3)[49]. The differential expression analysis was conducted by DESeq2 (version 1.22.2)[50]. Gene set enrichment analysis (GSEA) was performed by using clusterProfiler software (version 3.10.0) from the database of MsigDB (http://www.gsea-msigdb.org/). For metabolome analysis, GSEA tests the relative position of a collection of genes within an independent, ranked data set (Kyoto Encyclopedia of Genes and Genomes (KEGG)). The normalized enrichment score (NES) represents the number and differential expression intensity of the genes enriched in the corresponding cell subset.

**Single-cell RNA sequencing data analysis**. Drop sequencing was performed as described previously[22]. The sequencing reads were aligned to the reference mouse genome (mm10) and an expression matrix of read counts for each gene and cell was prepared using the Drop-seq software available from McCarroll Lab[22]. We employed the Seurat V3 software[51] for data integration and the core single-cell analysis. The resulting clusters were labelled based on the annotations given by SingleR[52]. Compositional analysis on the ratio data was performed by Dirichlet regression models. The trajectory with regard to B-cell differentiation was modelled by the Palantir algorithm[53].

**Statistics and reproducibility**. All data are represented as mean ± SEM, unless otherwise described. Comparisons between two samples were done using the unpaired or paired Student's t-tests. Statistical analyses were performed with GraphPad Prism 8 (MDF).

**Reporting summary**. Further information on research design is available in the Nature Portfolio Reporting Summary linked to this article.

### Data availability

Source data for the graphs and charts are available as Supplementary Data 1 and any remaining information can be obtained from the corresponding authors upon reasonable request. RNA sequence data are available at GEO under accession number GSE197720 and GSE197721.

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

## Acknowledgements

This study was supported by Grants-in-Aid for Scientific Research from the Japan Society for the Promotion of Science (grants 20H03711 and 23H02935 to A.F.; 21H02950 to Y.Kunisaki), and AMED under Grant Number JP19bm0404052h0001 to A.F. We would like to acknowledge all our colleagues in Dr. Kang's, Dr. Akashi's and Dr. Arai's laboratory for their support throughout this project. We appreciate the technical support from the Research Support Centre, Graduate School of Medical Sciences, Kyushu University, and from the Medical Institute of Bioregulation, Kyushu University.

## Author contributions

K.N., Y.Kunisaki and F.A. designed the study. K.N. and Y.Kunisaki carried out most of the experiments and analysed the data. K.H., H.Y., and R.Y. performed the in vitro differentiation assays of MSCs and K.G. conducted the experiments to evaluate mitochondrial functions. Y.S., J.N., Y. Kikushige, P.S.S. and B.D.M. performed the single-cell sequencing experiments and analysed the data. D.K., K.A., S.O. and F.A. supervised the experiments and reviewed the data. K.N. and Y.Kunisaki prepared the figures and K.N., Y.Kunisaki and F.A. wrote the manuscript.

## Competing interests

The authors declare no competing interests.
