## [Peer Review File · Communications Biology]

Reviewers' comments:

Reviewer #1 (Remarks to the Author):

Brief summary

Previous work by the authors has identified decreased POT1a production with age-associated myeloid bias: POT1a was found to be elevated in hematopoietic stem cells (HSCs) of young mice and decreased in old mice. These prior studies showed a role for POT1a in preventing myeloid skewing – a hematopoietic bias known to occur during normal aging – by protecting the HSCs from DNA damage responses and reactive oxygen species at the telomeres. The authors thus suggested that a similar mechanism would be important in maintaining other stem cells as well. The authors suggest here that POT1a is necessary for osteo-differentiation of MSCs.

Previous work by numerous groups have implicated mesenchymal stem cells (MSCs or mesenchymal stem/progenitor cells, MSPCs) in several aspects of supporting and regulating hematopoiesis within the adult bone marrow. Thus the authors seek here to identify a role for POT1a in MSCs with particular regard to the MSCs' regulation of hematopoiesis. To do this, the authors use a MSC-specific POT1a knockdown followed by single cell gene profiling of bone marrow cells to show that POT1a is crucial for MSC support of lymphopoiesis. This work thus further strengthens the authors' previous claim for a role of POT1a in maintaining lymphoid differentiation and the roll of loss of POT1a in age-related myeloid-skewing.

Overall Impression

The most important aspect of this manuscript is the involvement of POT1a in myeloid-skewing – a process that is poorly understood during normal aging.

Overall this manuscript seems like the authors are using their data to tell the story they want to tell, rather than telling the story that their data shows. This is particularly true for the role of POT1a in osteogenic differentiation: the data does not convince me that the functional changes are specific to this lineage so much as a loss of the stem cell state overall, as indicated to the change in immunomodulatory pathways and other differentiation lineages noted.

The introduction would benefit from reworking. It is missing key connections and requires some corrections as listed below.

The references must be checked and updated, especially within the introduction: many are old but not seminal papers (e.g. the review cited as reference 13). A noticeable percentage of references are incorrectly noted: some of these are indicated within the specific comments below.

Specific comments

I. Introduction -

1. Introduction: The introduction does not indicate why the authors are interested in looking at POT1a in MSCs with regards to hematopoiesis. The manuscript would strongly benefit from further introduction to the bone marrow niche with a brief overview of role of MSCs in regulating hematopoiesis; paragraph 4 begins to address this but does not offer sufficient explanation given the scope of this manuscript. For example see work from the laboratories of Francesco Dazzi, Paul

Frenette, and/or Pranela Rameshwar, among others – as the authors discuss later. There is extensive literature on MSCs in normal and malignant hematopoiesis; some work on malignant hematopoiesis has direct link to aging hematopoiesis. The authors begin to address this while discussing the first result: absence of mentioning this link makes it difficult for a non-hematologist or non-MSc expert to follow. Reference 8, which is cited in this vein, is not applicable. The authors should also consider addressing discrepancies between human and murine MSCs: human and murine MSCs are quite different and findings from one do not always translate to the other; but human MSCs can retain their function when xenotransplanted into a murine model.

2. Introduction Lines 81-83: The MSC secretome also plays a key role in this process

3. Introduction Lines 83-84: The authors write "MSPCs are typically isolated from bone marrow using a panel of surface markers or using reporter constructs." This is misleading. It is this reviewer's belief that MSCs that are to be expanded in the lab are usually isolated based on plastic adherence and confirmed based on surface markers. MSCs are a heterogeneous population of cells, thus isolating the MSCs by surface markers will either yield a subpopulation of MSCs (as was noted and done in this paper) or a non-pure population of MSCs.

3. Introduction Line 86: The authors should clarify what they mean by "MSC activity:" are they just referring to differentiation capacity, or are they referring to all MSC activity, including immunomodulatory activity, for example.

4. Introduction Line 87: References 9-11 seem misplaced.

II. Result 1: POT1a deletion in MSPCs leads to impaired osteo-lineage differentiation -

5. Results Line 112: If the authors are including discussion within their results section, it would be beneficial to note that PDGFR α + CD51+ MSCs are those that support HPCs

6. Figure 1d: An insert showing the correct morphology of these cells would be beneficial to ascertain that the observed results are due to the correct cell type. At this scale the morphology is unclear.

7. Figure 1g: The authors indicate that there is no significance in telomere length. In the absence of statistics this change looks significant. Statistics should be noted if this change is indeed not significant

8. Figure 1k/Line 141: The authors indicate that HSC-related genes are unchanged. Why would the authors expect HSC-related genes to be changed in MSCs? It would be much more relevant to assess genes known to be associated with multipotent MSCs (e.g. Oct4a, nanog, Sox2). In the following sentence the authors suggest that "reduced multipotency may be the primary effect of POT1a deletion." This appears to be the case, but is not the same as claiming that POT1a is needed for osteogenic differentiation.

9. Figure 1l. These images are taken from too low an objective to clearly note any changes or lack thereof (compare to Supp 1f).

III. Result 2: POT1a deficiency induces intracellular fatty acid accumulation in MSPCs -

10. Figure 2: These findings appear in line with the MSPCs losing their stemness and thus having been unable to differentiate towards an osteogenic (or chondrogenic) lineage. Unless the authors feel this to be conclusion to be incorrect, this should be indicated in the text.

IV. Result 3: MSPC-targeted POT1a-deficient mice exhibit impaired skeletal development -

11. The conclusion at the end of this section (lines 222-226) seems far reaching without an overexpression system. The data strongly suggest that the wild type contains cell capable of undergoing osteogenic differentiation, but it not does show that POT1a preferentially influences osteogenic differentiation versus any other lineage.

V. Result 4: Bone marrow microenvironments composed of POT1a-deleted MSCs impair B-cell development -

12. Inclusion of a CFU-GEMM or CFU-GM type assay versus a CFU-ProB, for example, would strengthen this data.

VI. Result 5: POT1a deficiency in MSCs impacts bone growth

13. There is insufficient data to support a conclusion such as is stated in lines 296-299.

VII. Discussion -

14. The aforementioned comments should be addressed within the discussion where appropriate.

VIII. Methods -

15. In vitro culture of MSCs: The following should be included within this section: what types of plates were used (e.g. vacuum gas plasma treated tissue culture plates), the confluency at which the cells were passaged, what passages of cells were used for experiments, how were the adherent cells collected (e.g. using trypsin EDTA), and culture conditions (e.g. 37C, 5% CO₂).

16. In vitro MSC differentiation assay: Were these medias used as per the manufacturer's recommendation? Was media changed at any point during these periods of differentiation?

17. Q-PCR: It is always beneficial to include a table of primer sequences, often within the supplemental files

18. Long term reconstitution assay: What dose of radiation was used? How were the mice irradiated?

Reviewer #2 (Remarks to the Author):

In this study, Nakashima and colleagues investigated the role of sheltering complex member Pot1a in mesenchymal stem cells (MSCs) biology. The authors show that Pot1a deficiency in the bone marrow MSCs leads to impaired osteoblastic differentiation in vitro and to skeletal retardation in vivo. Additionally, the authors linked decreased osteo-lineage commitment to an augmented fatty acid accumulation in the MSCs, which results in excessive ROS production and subsequent DNA damage. Moreover, single-cell transcriptome analysis of the bone marrow from mice with MSC-specific Pot1a deletion revealed ineffective B lymphopoiesis, one of the hallmarks of aging hematopoietic system. This study is very timely, considering growing interest in bone marrow microenvironment, especially in blood malignancies, and connection to telomeropathies caused by mutations in genes encoding sheltering complex components. The observations and profiling provided by the authors are interesting and embark on rather unexplored topic of telomer-associated protein complexes in MSC biology. However, the manuscript at the current form is descriptive and lacs deeper experimental validation to support statements made by the authors. Please see more specific comments below.

Major comments:

1. The connection between aging, telomers and decreased Pot1a level is not completely clear to the

reviewer. The authors nicely show lower expression of osteo-lineage genes in aged MSCs (rather expected result), no changes in a telomer length in aged MSCs but lower Pot1a expression. How does Pot1a regulate the osteo-lineage genes then (is it a cause or a consequence of diminished osteoblastic gene expression and aging process)? Would overexpression of Pot1a in aged MSCs recover osteo-lineage gene expression?

2. The authors show increased expression of genes involved in fatty acids accumulation. What about genes regulating β -oxidation of fatty acids (Cpt1, Acad, Echs1, Hadh, Acaa2 etc.), are they decreased? Related to this observation, could authors comment on the fact that increased expression of genes regulating fatty acids accumulation (Fig. 2b) is observed mainly in the MSCs cultured for 4 weeks; is it a combined effect of Pot1a deficiency and "aging"?

3. According to the authors, the mechanism of impaired osteoblastic differentiation involves ROS and DNA damage. Rescue experiments that can be performed using in vitro cultures of MSCs would definitively strengthen this claim. Treating the MSCs with known ROS scavengers (NAC) or expressing ROS scavenging enzymes to alleviate the oxidative stress and decrease DNA damage to improve osteoblastic differentiation could be considered. Alternatively, blocking fatty acid accumulation by CD36 neutralization or downregulation of Pparg etc. might be an experimental option to explore.

4. The authors hypothesize that reduced IL-7 is responsible for impaired B lymphopoiesis. However, there are no differences in IL-7 expression between WT and Pot1a KO MSCs or BLCs. Hence, this hypothesis seems unsubstantiated to the reviewer. Measurement of IL-7 in the conditioned media from WT vs. Pot1a KO MSCs should be considered. Does culture of CLPs or pro-B cells from Pot1a mutant mice in the presence of recombinant IL-7 alleviate the differentiation block?

Minor comments:

1. Please provide more B cell marker genes (Ebf1, Pax5, IL-7R etc.) in Fig. 4C and representative FACS plots for Fig. 4F.

2. Are differences in granulocytes and monocytes in Fig. 4B and neutrophil ration in Fig. 4H statistically significant? The same question for telomer length in Fig. 1G.

3. The authors should consider rephrasing some statements in the discussion about importance of telomere protection for MSC stemness (lines 350-353 and 378-380) since they clearly show no differences in telomer length between WT and Pot1a deficient MSCs, neither they really tested the MSC stemness but rather MSC differentiation potential, and the Pot1a-dependent increase in DNA damage is currently just an observation and has no mechanistic link to MSC stemness.

Reviewer #3 (Remarks to the Author):

The manuscript submitted by Nakashima et al contains exciting results and is in overall well written. However, I do have a few concerns and suggestions:

Perhaps the major weakness of this manuscript is the assessment of MSPC function in vivo is based on NSG2-transgenic mice, where NG2 is a commonly expressed in all pericytes throughout the body. In consequence, the observed results can hardly be attributed to defects in bone marrow MSPCs. To overcome this major conceptual problem, I recommend acknowledging that POT1a depletion in NG2-Cre mice will affect all pericytes, where in the bone marrow, at least a fraction of MSPCs correspond with pericytes. Consequently, the text should be rewritten to avoid confusion. For example, the subtitle "MSPC-targeted POT1a-deficient mice exhibit impaired skeletal development" should read "POT1a-deficiency in NG2 positive cells causes impaired skeletal development".

Other concerns:

The concept of MSPC is somewhat confusing, although this is an acknowledged subject of discussion in the field. Pericytes, skeletal stem cells, bone marrow stroma, and MSCs, although likely related cell types, also show remarkable differences. To avoid adding confusion by introducing the rather novel concept of MSPCs, I recommend describing the cells as simply MSC (standing for both, mesenchymal stem cells and multipotent stromal cells). The exact identity of these cells should be clear based on the tissue of origin (bone marrow) and the markers used for isolation (PDGFR α and CD51 in vitro and NG2 in vivo).

In the introduction (P3L69), it says "We recently reported that...", I suggest changing it to "We previously reported that...", since the publication is now 5 years old.

P4L77 please avoid mentioning the adipose tissue as in this context it could be misleading: this is not a site of hematopoiesis. In fact, to be precise, MSC-like cells derived from perivascular cells (pericytes and adventitial stromal cells) can be found in virtually any vascularized tissue.

P4L89 please change to "...loss of integrin beta3 signaling", for clarity.

P5L105-108, these sentences are very similar to P4L75-76. Please edit to avoid duplications.

P7L146 please state "...osteogenic, adipogenic, etc", osteo/adipo/chondro is laboratory jargon.

P7L149, please replace for "These results suggest that..." because the described results are in vitro and the statement tries to be more general than that.

Figure 1e is somehow confusing. Is an $n = 1$ shown? Please include biological replicates (either averaged or not, if necessary, as supplementary information). It is surprising that despite the reported effect of PAT1a on mitochondrial function, no effects are seen at proliferation rates. Please consider commenting on this in the discussion. Potential additional experiments would include measuring glucose consumption and lactate secretion.

In Figure 2C, if any statistics were applied to these results, please include the appropriate notations (*). Also, ideally the lines could be drawn thinner, to better appreciate the differences in between conditions.

P11L257. flow cytometry (two words).

P12L265. Do the authors mean: "in the bone marrow, tdTomato signals were exclusively observed in CD45-...."?

P12L277. As a precaution (and in favor of accuracy), I recommend "...the findings observed in the bone sections (Fig. 3i) are likely attributed to defects in differentiation potential of POT1a-deficient MSCs".

P12L279. To characterize (z, no s).

P12L283 please spell out GSEA.

The conclusions based on Figure 5 are confusing. IL-7 are not lower in the sorted cells. The authors need to either rewrite this section or perform additional experiments to demonstrate low IL-7 levels in NS2-cre/loxp-tdTomato/Pot1a(fl/fl) mice. Even more, rescue experiments with supplementation of IL-7 are also necessary to support the conclusions as written.

Dear reviewers,

Below are our point-by-point answers to the reviewers' comments based on the results we have obtained. Corrections in the main text are written in red.

Reviewers' comments:

Reviewer #1 (Remarks to the Author):

Brief summary

Previous work by the authors has identified decreased POT1a production with age-associated myeloid bias: POT1a was found to be elevated in hematopoietic stem cells (HSCs) of young mice and decreased in old mice. These prior studies showed a role for POT1a in preventing myeloid skewing – a hematopoietic bias known to occur during normal aging – by protecting the HSCs from DNA damage responses and reactive oxygen species at the telomeres. The authors thus suggested that a similar mechanism would be important in maintaining other stem cells as well. The authors suggest here that POT1a is necessary for osteo-differentiation of MSCs.

Previous work by numerous groups have implicated mesenchymal stem cells (MSCs or mesenchymal stem/progenitor cells, MSPCs) in several aspects of supporting and regulating hematopoiesis within the adult bone marrow. Thus the authors seek here to identify a role for POT1a in MSCs with particular regard to the MSCs' regulation of hematopoiesis. To do this, the authors use a MSC-specific POT1a knockdown followed by single cell gene profiling of bone marrow cells to show that POT1a is crucial for MSC support of lymphopoiesis. This work thus further strengthens the authors' previous claim for a role of POT1a in maintaining lymphoid differentiation and the roll of loss of POT1a in age-related myeloid-skewing.

Overall Impression

The most important aspect of this manuscript is the involvement of POT1a in myeloid-skewing – a process that is poorly understood during normal aging.

Overall this manuscript seems like the authors are using their data to tell the story they

want to tell, rather than telling the story that their data shows. This is particularly true for the role of POT1a in osteogenic differentiation: the data does not convince me that the functional changes are specific to this lineage so much as a loss of the stem cell state overall, as indicated to the change in immunomodulatory pathways and other differentiation lineages noted.

The introduction would benefit from reworking. It is missing key connections and requires some corrections as listed below.

The references must be checked and updated, especially within the introduction: many are old but not seminal papers (e.g. the review cited as reference 13). A noticeable percentage of references are incorrectly noted: some of these are indicated within the specific comments below.

Specific comments

I. Introduction -

1. Introduction: The introduction does not indicate why the authors are interested in looking at POT1a in MSCs with regards to hematopoiesis. The manuscript would strongly benefit from further introduction to the bone marrow niche with a brief overview of role of MSCs in regulating hematopoiesis; paragraph 4 begins to address this but does not offer sufficient explanation given the scope of this manuscript. For example see work from the laboratories of Francesco Dazzi, Paul Frenette, and/or Pranela Rameshwar, among others – as the authors discuss later. There is extensive literature on MSCs in normal and malignant hematopoiesis; some work on malignant hematopoiesis has direct link to aging hematopoiesis. The authors begin to address this while discussing the first result: absence of mentioning this link makes it difficult for a non-hematologist or non-MSc expert to follow. Reference 8, which is cited in this vein, is not applicable. The authors should also consider addressing discrepancies between human and murine MSCs: human and murine MSCs are quite different and findings from one do not always translate to the other; but human MSCs can retain their function when xenotransplanted into a murine model.

In addition to Paul Frenetti's paper as reference 13-16, we cited 3 other papers (reference 9-11), including Francesco Dazzi's papers (reference 10-11) in paragraph 3 and added the text as "Although human MSCs share the characteristics with murine ones, they are isolated with some unique surface markers other than the orthologs. In addition, human MSCs have been elucidated to play roles in various pathologies due to their predisposition

to engraft in xenograft mouse models⁹⁻¹¹. MSCs are a heterogeneous subset of stromal stem cells that can be isolated from many tissues. Some of the subpopulations of MSCs can be isolated from bone marrow using a panel of surface markers.” In lines 81-87. In addition, we have cited the paper published by Pranela Rameshwar et al. reporting the association between aged MSCs and hematopoietic malignancy as reference 18 and have revised the text as “Several studies show that microenvironments composed of aged MSCs associate with leukemia development¹⁸” in lines 100-102. We apologize the inappropriate citation of reference 8 and replaced it to the appropriate paper as reference 12.

2. Introduction Lines 81-83: The MSC secretome also plays a key role in this process Thank you for your suggestion. We added the text and the reference about the MSC secretome in lines 80-81 as “...and exhibit anti-inflammatory effects by secreting cytokines.”

3. Introduction Lines 83-84: The authors write “MSPCs are typically isolated from bone marrow using a panel of surface markers or using reporter constructs.” This is misleading. It is this reviewer’s belief that MSCs that are to be expanded in the lab are usually isolated based on plastic adherence and confirmed based on surface markers. MSCs are a heterogeneous population of cells, thus isolating the MSCs by surface markers will either yield a subpopulation of MSCs (as was noted and done in this paper) or a non-pure population of MSCs.

We agree this reviewer’s suggestion and rewrote this part in lines 84-87 as “MSCs are a heterogeneous subset of stromal stem cells that can be isolated from many tissues. Some of the subpopulation of MSCs can be isolated from bone marrow using a panel of surface markers.”

3. Introduction Line 86: The authors should clarify what they mean by “MSC activity:” are they just referring to differentiation capacity, or are they referring to all MSC activity, including immunomodulatory activity, for example.

As the reviewer pointed out, this “MSC activity” indicated just differentiation capacity. We changed “MSC activity” to “MSC’s differentiation capacity” in line 92.

4. Introduction Line 87: References 9-11 seem misplaced.

Thank you for your suggestion. This part describes the relationship between Nestin-GFP transgenic mice and NG2-cre mice, which is quoted from reference 9 and 10. Therefore we deleted reference 11 in this part. (In the current version, ref.9 and 10 are changed to ref.13 and 14 respectively.)

II. Result 1: POT1a deletion in MSCs leads to impaired osteo-lineage differentiation -

5. Results Line 112: If the authors are including discussion within their results section, it would be beneficial to note that PDGFR α ⁺ CD51⁺ MSCs are those that support HPCs
Having this helpful comment, we have added a text to define PDGFR α ⁺ CD51⁺ MSCs as “...we analysed the expression of genes associated with osteogenic-, adipogenic- and chondrogenic- differentiation in the bone marrow PDGFR α ⁺ CD51⁺ MSCs, which mark human Nestin+ sphere-forming MSCs capable of hematopoietic progenitor cell expansion, sorted from 8-10 weeks old adult and ~120 weeks old aged mice.” in lines 117-120.

6. Figure 1d: An insert showing the correct morphology of these cells would be beneficial to ascertain that the observed results are due to the correct cell type. At this scale the morphology is unclear.

Thank you for your suggestion. We added high-power-field photos.

7. Figure 1g: The authors indicate that there is no significance in telomere length. In the absence of statistics this change looks significant. Statistics should be noted if this change is indeed not significant

Thank you for your suggestion. The p-value for this experiment was 0.104, which was not significant. We added this result in the figure.

8. Figure 1k/Line 141: The authors indicate that HSC-related genes are unchanged. Why would the authors expect HSC-related genes to be changed in MSCs? It would be much more relevant to assess genes known to be associated with multipotent MSCs (e.g. Oct4a, nanog, Sox2). In the following sentence the authors suggest that “reduced multipotency may be the primary effect of POT1a deletion.” This appears to be the case, but is not the same as claiming that POT1a is needed for osteogenic differentiation.

The genes shown in Supplementary Fig. 1e are related to "HSC maintenance," not "HSC-related." We agree that the term "HSC maintenance" is misleading. Therefore, we have rephrased "genes related to HSC maintenance" to "genes encoding factors to support HSC activity (HSC niche-related)" in lines 151-152. In addition, we also got into the genes associated with multipotency (e.g., *Oct4a*, *Nanog*, *Sox2*) and added the results in lines 152-153 and Supplementary Fig. 1f.

9. Figure 1l. These images are taken from too low an objective to clearly note any changes or lack thereof (compare to Supp 1f).

Thank you for your suggestion. We added high-power-field photos.

III. Result 2: POT1a deficiency induces intracellular fatty acid accumulation in MSCs -

10. Figure 2: These findings appear in line with the MSCs losing their stemness and thus having been unable to differentiate towards an osteogenic (or chondrogenic) lineage. Unless the authors feel this to be conclusion to be incorrect, this should be indicated in the text.

We thank this reviewer's suggestion. We've added the sentence "..., suggesting ROS and consequent DNA damage at least in part contribute to the impaired osteoblast differentiation in POT1a deleted MSCs (Fig.2h). These findings indicate that POT1a-deficient MSCs lose osteogenic differentiation potency through ROS production." in lines 204-207.

IV. Result 3: MSC-targeted POT1a-deficient mice exhibit impaired skeletal development -

11. The conclusion at the end of this section (lines 222-226) seems far reaching without an overexpression system. The data strongly suggest that the wild type contains cell capable of undergoing osteogenic differentiation, but it not does show that POT1a preferentially influences osteogenic differentiation versus any other lineage.

We thank the reviewer for this suggestion, which we agree is important to show the association between POT1a and the aged phenotype of MSCs. To this end, we have performed experiments in which MSCs were transfected with a retroviral vector to overexpress Pot1a in culture. The rescue with POT1a in MSCs isolated from the old mice has restored the expression of several genes related to osteogenic differentiation, such as *Spp1*, *Alpl* and *Col1a1*, suggesting the importance of Pot1a for MSCs to sustain multi-potency. These results are shown in Figure 1a in the revised version, and we added the text in lines 128-132 as below. "Moreover, we have performed experiments, in which MSCs were transfected with a retroviral vector to overexpress Pot1a in culture. The rescue with Pot1a in MSCs isolated from the old mice has restored the expression of genes related to osteogenesis, such as *Alpl*, *Col1a1*, and *Opn*, suggesting the importance of POT1a for MSCs to sustain multi-potency (Fig. 1a)."

V. Result 4: Bone marrow microenvironments composed of POT1a-deleted MSCs impair B-cell development -

12. Inclusion of a CFU-GEMM or CFU-GM type assay versus a CFU-ProB, for example, would strengthen this data.

We agree that the experiments suggested by this reviewer are essential. We performed CFU assays with the bone marrow cells from the NG2-cre/Pot1a^{fl/fl} mice. The bone marrow

cells from the NG2-cre/Pot1a^{fl/fl} mice gave rise to smaller sizes of CFU-PreB colonies and their numbers were significantly reduced compared to the controls whereas CFU-GM colonies were comparably observed. We added these results in the main text in lines 281-284 (Fig. 4j and Supplementary Fig. 4k).

VI. Result 5: POT1a deficiency in MSCs impacts bone growth

13. There is insufficient data to support a conclusion such as is stated in lines 296-299. In the revised version, we have re-written this part as "reduced number of bone lining osteoblasts" in the main text in line 317.

VII. Discussion -

14. The aforementioned comments should be addressed within the discussion where appropriate.

We only speculate that the decrease in the number of osteoblasts may contribute to the decrease in IL-7, and we have already mentioned it in the discussion part in lines 380-391.

VIII. Methods -

15. In vitro culture of MSCs: The following should be included within this section: what types of plates were used (e.g. vacuum gas plasma treated tissue culture plates), the confluency at which the cells were passaged, what passages of cells were used for experiments, how were the adherent cells collected (e.g. using trypsin EDTA), and culture conditions (e.g. 37°C, 5% CO₂).

Types of plate: vacuum gas plasma treated tissue culture plates. The confluency at which the cells were passaged: 100% confluency. What passages of cells were used for experiments: 3 times during culture (once per week). How were the adherent cells collected: using Accutase. Culture conditions: 37°C, 5% O₂ and 10% CO₂ incubator. We added this information into Method part.

16. In vitro MSC differentiation assay: Were these medias used as per the manufacturer's recommendation? Was media changed at any point during these periods of differentiation? We follow the product instruction. Osteogenic differentiation: once per 3-4 days; Adipocyte differentiation: once per 2-3 days; Chondrogenic differentiation: once per 3-4 days.

17. Q-PCR: It is always beneficial to include a table of primer sequences, often within the supplemental files

We used TaqMan probe at the Q-PCR examination. We have added probe information table in the supplementary table.

18. Long term reconstitution assay: What dose of radiation was used? How were the mice irradiated?

We apologize for the lack of description. The mice were irradiated 9.5Gy and we have added the information in the Methods section.

Reviewer #2 (Remarks to the Author):

In this study, Nakashima and colleagues investigated the role of sheltering complex member Pot1a in mesenchymal stem cells (MSCs) biology. The authors show that Pot1a deficiency in the bone marrow MSCs leads to impaired osteoblastic differentiation in vitro and to skeletal retardation in vivo. Additionally, the authors linked decreased osteo-lineage commitment to an augmented fatty acid accumulation in the MSCs, which results in excessive ROS production and subsequent DNA damage. Moreover, single-cell transcriptome analysis of the bone marrow from mice with MSC-specific Pot1a deletion revealed ineffective B lymphopoiesis, one of the hallmarks of aging hematopoietic system. This study is very timely, considering growing interest in bone marrow microenvironment, especially in blood malignancies, and connection to telomeropathies caused by mutations in genes encoding sheltering complex components. The observations and profiling provided by the authors are interesting and embark on rather unexplored topic of telomer-associated protein complexes in MSC biology. However, the manuscript at the current form is descriptive and lacks deeper experimental validation to support statements made by the authors. Please see more specific comments below.

Major comments:

1. The connection between aging, telomers and decreased Pot1a level is not completely clear to the reviewer. The authors nicely show lower expression of osteo-lineage genes in aged MSCs (rather expected result), no changes in a telomer length in aged MSCs but lower Pot1a expression. How does Pot1a regulate the osteo-lineage genes then (is it a cause or a consequence of diminished osteoblastic gene expression and aging process)? Would overexpression of Pot1a in aged MSCs recover osteo-lineage gene expression?

We thank the reviewer for this suggestion, which we agree is important to show the association between POT1a and the aged-phenotype of MSCs. To this end, we have performed experiments in which MSCs were transfected with a retroviral vector to overexpress Pot1a in culture. The rescue with POT1a in MSCs isolated from the old mice has restored the expression of several genes related to osteogenic differentiation, such as *Spp1*, *Alpl* and *Col1a1*, suggesting the importance of Pot1a for MSCs to sustain multi-potency. These results are shown in Figure 1a in the revised version and we added the text in lines 128-132 as below. "Moreover, we have performed experiments, in which MSCs were transfected with a retroviral vector to overexpress Pot1a in culture. The rescue with Pot1a in MSCs isolated from the old mice has restored the expression of genes related to osteogenesis such as *Alpl*, *Col1a1*, and *Opn*, suggesting the importance of POT1a for MSCs to sustain multi-potency (Fig. 1a)."

2. The authors show increased expression of genes involved in fatty acids accumulation. What about genes regulating β -oxidation of fatty acids (Cpt1, Acad, Echs1, Hadh, Acaa2 etc.), are they decreased? Related to this observation, could authors comment on the fact that increased expression of genes regulating fatty acids accumulation (Fig. 2b) is observed mainly in the MSCs cultured for 4 weeks; is it a combined effect of Pot1a deficiency and “aging”?

We've added the graph comparing the genes regulating β -oxidation of fatty acids between WT and Pot1a deficient MSCs in Supplementary Figure 2c, which show that the expression of genes related to β -oxidation was not decreased. We also have added a sentence describing these results in the text in lines 186-188.

We added our speculation about this in lines 343-350 in the discussion part as “POT1a deficiency in MSCs appears to affect differentiation potential rather than cell proliferation, arguing that cell differentiation requires additional metabolic activation while steady cell growth can be achieved with baseline oxygen consumption as reported with HSC differentiation^{30,31}. In our experiments, there was no significant effect on POT1a deficient MSCs cellular growth in vitro despite mitochondrial dysfunction and decreased FAO. It is presumed that nucleotide synthesis is compensated by alternative pathways we have not identified in this study. “

3. According to the authors, the mechanism of impaired osteoblastic differentiation involves ROS and DNA damage. Rescue experiments that can be performed using in vitro cultures of MSCs would definitively strengthen this claim. Treating the MSCs with known ROS scavengers (NAC) or expressing ROS scavenging enzymes to alleviate the oxidative stress and decrease DNA damage to improve osteoblastic differentiation could be considered. Alternatively, blocking fatty acid accumulation by CD36 neutralization or downregulation of Pparg etc. might be an experimental option to explore.

We thank the reviewer's suggestion. To address this point, we have performed rescue experiments using an anti-oxidative agent, N-acetyl cysteine (NAC). Supplemented with NAC in culture media, the expression of genes related to osteogenesis *Alpl* and *Spp1* was partially restored, suggesting ROS and consequent DNA damage at least partly contribute to the impaired osteoblast differentiation in Pot1a deleted MSCs. These results are shown in Figure 2h and described in the main text in lines 199-207.

4. The authors hypothesize that reduced IL-7 is responsible for impaired B lymphopoiesis. However, there are no differences in IL-7 expression between WT and Pot1a KO MSCs or BLCs. Hence, this hypothesis seems unsubstantiated to the reviewer. Measurement of IL-7 in the conditioned media from WT vs. Pot1a KO MSCs should be considered. Does culture

of CLPs or pro-B cells from Pot1a mutant mice in the presence of recombinant IL-7 alleviate the differentiation block?

We agree with this point that the conclusion was misinterpreted. In the revised version, we have re-written this part as "reduced number of bone lining osteoblasts" in the main text in line 317.

Minor comments:

1. Please provide more B cell marker genes (*Ebf1*, *Pax5*, IL-7R etc.) in Fig. 4C and representative FACS plots for Fig. 4F.

We added the figures of *Ebf1*, *Pax5*, *Il-7r* in Fig.4c. The representative FACS plot for pre pro B cell in Supplementary Fig.4j.

2. Are differences in granulocytes and monocytes in Fig. 4B and neutrophil ration in Fig. 4H statistically significant? The same question for telomere length in Fig. 1G.

The p-value for the count of granulocytes and monocytes, neutrophil ration and telomere length was 0.510, 0.881, 0.087, and 0.104, which were not significant. We added these results in Fig. 1g and Fig. 4b.

3. The authors should consider rephrasing some statements in the discussion about importance of telomere protection for MSC stemness (lines 350-353 and 378-380) since they clearly show no differences in telomere length between WT and Pot1a deficient MSCs, neither they really tested the MSC stemness but rather MSC differentiation potential, and the Pot1a-dependent increase in DNA damage is currently just an observation and has no mechanistic link to MSC stemness.

Thank you for your suggestion. As you pointed out, we do not show the data about MSC stemness. We rephrased "for maintenance of the osteogenic- differentiation potential" in lines 378-379, and deleted the last paragraph in the discussion part.

Reviewer #3 (Remarks to the Author):

The manuscript submitted by Nakashima et al contains exciting results and is in overall well written. However, I do have a few concerns and suggestions:

Perhaps the major weakness of this manuscript is the assessment of MSPC function in vivo is based on NSG2-transgenic mice, where NG2 is a commonly expressed in all pericytes throughout the body. In consequence, the observed results can hardly be attributed to defects in bone marrow MSCs. To overcome this major conceptual problem, I recommend acknowledging that POT1a depletion in NG2-Cre mice will affect all pericytes, where in the bone marrow, at least a fraction of MSCs correspond with pericytes. Consequently, the text should be rewritten to avoid confusion. For example, the subtitle “MSC-targeted POT1a-deficient mice exhibit impaired skeletal development” should read “POT1a-deficiency in NG2 positive cells causes impaired skeletal development”.

We appreciate the suggestion. We have revised the text in lines 212-213 as “a NG2-cre strain (hereafter NG2-cre/Pot1a^{fl/fl}) that targets all pericytes in whole body and marks ~96% of Nestin-GFP positive MSCs in BM”.

We changed the subtitle as you suggested in lines 209-210.

Other concerns:

The concept of MSPC is somewhat confusing, although this is an acknowledged subject of discussion in the field. Pericytes, skeletal stem cells, bone marrow stroma, and MSCs, although likely related cell types, also show remarkable differences. To avoid adding confusion by introducing the rather novel concept of MSCs, I recommend describing the cells as simply MSC (standing for both, mesenchymal stem cells and multipotent stromal cells). The exact identity of these cells should be clear based on the tissue of origin (bone marrow) and the markers used for isolation (PDGFRa and CD51 in vitro and NG2 in vivo). In the introduction (P3L69), it says “We recently reported that...”, I suggest changing it to “We previously reported that...”, since the publication is now 5 years old.

According to this reviewer’s suggestion, we have re-written the term MSCs to mesenchymal stem cells and multipotent stromal cells (MSCs) in the whole manuscript.

P4L77 please avoid mentioning the adipose tissue as in this context it could be misleading: this is not a site of hematopoiesis. In fact, to be precise, MSC-like cells derived from perivascular cells (pericytes and adventitial stromal cells) can be found in virtually any vascularized tissue.

Thank you for your suggestion. We agree that it's misleading. We deleted "adipose tissue" in line 77.

P4L89 please change to "...loss of integrin beta3 signalling", for clarity.

This is not "integrin beta 3" but "adrenergic β 3". Adrenergic nerve in bone marrow associates with the arteriole and hematopoietic stem cell niche(ref.12). We changed here to "loss of β 3 adrenergic receptor signalling" in line 95.

P5L105-108, these sentences are very similar to P4L75-76. Please edit to avoid duplications.

Thank you for this reviewer's advice, which we agree. We have deleted P5L105-108 and ref.13.

P7L146 please state "...osteogenic, adipogenic, etc", osteo/adipo/chondro is laboratory jargon.

Thank you for this reviewer's suggestion. We've changed as you pointed out.

P7L149, please replace for "These results suggest that..." because the described results are in vitro and the statement tries to be more general than that.

Thank you for your comment. We've edited it as you pointed out in line 153.

Figure 1e is somehow confusing. Is an n = 1 shown? Please include biological replicates (either averaged or not, if necessary, as supplementary information). It is surprising that despite the reported effect of POT1a on mitochondrial function, no effects are seen at proliferation rates. Please consider commenting on this in the discussion. Potential additional experiments would include measuring glucose consumption and lactate secretion.

Figure 1e showed representative data (n=1). We re-analyzed the data including three replicated experiments. The graph in the revised Figure 1e represents the average of the 3 experiments and shows no significant difference.

We added our speculation about this in lines 343-350 in the discussion part as "POT1a deficiency in MSCs appears to affect differentiation potential rather than cell proliferation, arguing that cell differentiation requires additional metabolic activation while steady cell growth can be achieved with baseline oxygen consumption as reported with HSC differentiation^{30,31}. In our experiments, there was no significant effect on Pot1a deficient MSCs cellular growth in vitro despite mitochondrial dysfunction and decreased FAO. It is presumed that nucleotide synthesis is compensated by alternative pathways we have not identified in this study."

In Figure 2C, if any statistics were applied to these results, please include the appropriate notations (*). Also, ideally the lines could be drawn thinner, to better appreciate the differences in between conditions.

Thank you for your suggestion. We thinned the lines, and about notations mark, we indicate it in the right bar graphs about OCR max and OCR spare in Figure 2c (middle and right panels).

P11 L257. flow cytometry (two words).

Thank you for your suggestion. We've edited it as you pointed out.

P12 L265. Do the authors mean: "in the bone marrow, tdTomato signals were exclusively observed in CD45-...."?

Yes, we do. tdTomato signals were almost exclusively observed in CD45-Ter119-CD31-non-hematopoietic and non-vascular cells.

P12 L277. As a precaution (and in favor of accuracy), I recommend "...the findings observed in the bone sections (Fig. 3i) are likely attributed to defects in differentiation potential of POT1a-deficient MSCs".

Thank you for your suggestion. We've changed as you pointed out in lines 297-298.

P12 L279. To characterize (z, no s).

Thank you for your suggestion. We've changed as you pointed out.

P12 L283, please spell out GSEA.

We've added GSEA's full spell "Gene Set Enrichment Analysis" in lines 144.

The conclusions based on Figure 5 are confusing. IL-7 are not lower in the sorted cells. The authors need to either rewrite this section or perform additional experiments to demonstrate low IL-7 levels in NG2-cre/loxp-tdTomato/Pot1a(fl/fl) mice. Even more, rescue experiments with supplementation of IL-7 are also necessary to support the conclusions as written.

We agree this point that the conclusion was misinterpreted. In the revised version, we have re-written this part as "reduced number of bone lining osteoblasts" in the main text in line 317.

REVIEWERS' COMMENTS:

Reviewer #1 (Remarks to the Author):

The authors have addressed each of my concerns. It is this reviewer's opinion that their manuscript regarding the role of Pot1a in MSCs for regulation of hematopoiesis is appropriate for publication.

Reviewer #2 (Remarks to the Author):

The author addressed all my comments and concerns and further improved the manuscript.

Reviewer #3 (Remarks to the Author):

The authors have addressed by comments and suggestions satisfactorily, thank you.

Dear reviewers,

Reviewers' comments:

Reviewer #1 (Remarks to the Author):

The authors have addressed each of my concerns. It is this reviewer's opinion that their manuscript regarding the role of Pot1a in MSCs for regulation of hematopoiesis is appropriate for publication.

We appreciate the effort in evaluating the updated manuscript. We are pleased that the modifications have adequately addressed the concerns raised by Reviewer #1.

Reviewer #2 (Remarks to the Author):

The author addressed all my comments and concerns and further improved the manuscript.

We appreciate your time and effort in evaluating the revised manuscript. We are pleased that the revisions have adequately addressed the concerns identified by Reviewer #2.

Reviewer #3 (Remarks to the Author):

The authors have addressed by comments and suggestions satisfactorily, thank you.

We are pleased to see that the updated manuscript has addressed the concerns of Reviewer #3.